# Bayesian target optimisation for high-precision holographic optogenetics

**Marcus A. Triplett**[1,2,†]   **Marta Gajowa**[3]   **Hillel Adesnik**[3]   **Liam Paninski**[1,2]

[1]Department of Statistics, Columbia University
[2]Zuckerman Mind Brain Behavior Institute, Columbia University
[3]Department of Molecular and Cell Biology, UC Berkeley
[†] `marcus.triplett@columbia.edu`

## Abstract

Two-photon optogenetics has transformed our ability to probe the structure and function of neural circuits. However, achieving precise optogenetic control of neural ensemble activity has remained fundamentally constrained by the problem of off-target stimulation (OTS): the inadvertent activation of nearby non-target neurons due to imperfect confinement of light onto target neurons. Here we propose a novel computational approach to this problem called Bayesian target optimisation. Our approach uses nonparametric Bayesian inference to model neural responses to optogenetic stimulation, and then optimises the laser powers and optical target locations needed to achieve a desired activity pattern with minimal OTS. We validate our approach in simulations and using data from *in vitro* experiments, showing that Bayesian target optimisation considerably reduces OTS across all conditions we test. Together, these results establish our ability to overcome OTS, enabling optogenetic stimulation with substantially improved precision.

## 1   Introduction

A key technological goal in neuroscience is to gain precise control over the activity of neurons. Currently, one of the most promising approaches for achieving such control is two-photon optogenetics [1–4]. This technique relies on the use of two-photon excitation to activate opsin molecules expressed in the somatic membrane, allowing individual neurons to be targeted for stimulation. Holographic optogenetics further extends this technique by focusing two-photon excitation into $\sim 10$ $\mu$m disks of light that illuminate all of a neuron's opsin molecules in parallel [5–9]. Replicating this illumination pattern across many neurons at once then grants simultaneous optogenetic control of entire neural ensembles [10–12]. However, the spatial precision of holographic optogenetics has remained fundamentally limited by the problem of off-target stimulation (OTS): if multiple opsin-expressing neurons lie in close proximity (in the most extreme case, if their membranes are in physical contact), two-photon excitation will frequently activate opsin molecules in each neuron when attempting to stimulate only one. This can inadvertently elicit spikes in non-target neurons, compromising the precision and specificity of the optogenetic manipulation [13].

Previous attempts at overcoming OTS have been either optical or molecular in nature (e.g. by using temporal focusing [2, 6, 7, 14] or improving the soma-targeting of the opsin [15, 16, 10]), but have not yet achieved "true" single-cell precision under many experimental conditions. Frequently, experimenters will simply express opsin sparsely in a population to avoid OTS, as this reduces the number of nearby opsin-expressing neurons that could be mistakenly activated [17]. However, sparse opsin expression reduces the number of neurons that can potentially be probed during an

experiment, and therefore limits the kinds of scientific questions that can be addressed using two-photon optogenetics. Conversely, in an ideal experiment, one could precisely stimulate any neuron despite a moderate or high density of opsin-expressing cells.

Here, we develop a novel computational strategy for all-optical experiments [18–21] that reduces (and in some cases entirely eliminates) OTS (Figure 1a, b). The key insight is that heterogeneity in opsin expression and intrinsic excitability causes neurons to have different sensitivities to optical stimulation. This implies that in some cases laser power could be safely turned down to prevent the spiking of a nearby non-target neuron that is weakly driven by stimulation while still spiking a target neuron that is strongly driven by stimulation. Further, the spatial arrangement of neurons can be exploited by moving a holographic target off of a neuron's nucleus (the typical stimulation point) towards a less dense area (e.g. towards neuropil or less photosensitive neurons) while still exciting a sufficient number of the target neuron's opsin molecules to elicit a spike. While some prior experimental research has considered cell-specific tuning of laser powers [22], our advance is to identify both the optimal laser powers and target locations automatically using efficient Bayesian techniques.

To achieve highly precise optogenetic ensemble control, we propose a two-phase process called Bayesian target optimisation (Bataro). In the first phase, we map the "optogenetic receptive field" (ORF) of each neuron by stimulating at various locations around cell nuclei and at a range of laser powers to test whether they elicit spikes. To infer these ORFs without exhaustively testing every combination of location and power, we use Gaussian processes (GPs) to encode prior knowledge about how neurons respond to optogenetic stimulation. Further, we propose to use holographic stimulation to test multiple locations around one or many neurons simultaneously, ensuring that the mapping phase completes quickly. This also allows us to model how neurons integrate two-photon excitation from multiple nearby holograms. In the second phase, we then exploit knowledge of the ORFs to optimise holographic targets. We use properties of the GP to infer gradients of an objective function at a spatial resolution that exceeds the original sampling resolution, enabling us to optimise for the exact holographic targets and laser powers needed to evoke a specific neural activity pattern.

## 2 Related work

Our work is related to prior research in three areas: GP-based receptive field inference, optimal stimulus design, and statistical methods for estimating connectivity using optogenetic stimulation. Since ref. [23], many studies have used GPs to infer the relationship between neural spiking and multi-dimensional covariates. This includes using GPs to infer place fields [24, 25] and orientation preference maps [26], as well as with Bayesian active learning techniques for inferring sensory receptive fields [27, 28]. However, none to our knowledge have used GP inference to model responses to optogenetic stimulation. Closed-loop methods for optimal control of neural activity using one-photon or electrical stimulation have been recently explored [29–34], though these approaches lack the necessary spatial specificity for highly precise ensemble control. Ref. [35] considers optimising spike-timing using two-photon stimulation (in addition to electrical stimulation), but focused on single neurons and did not use GP estimation methods. Finally, a number of papers have developed statistical models of optogenetic data to infer functional or synaptic connectivity [36–43], but do not consider identifying the exact stimulation parameters needed to evoke specific activity patterns. Here, we unify these three approaches to develop a novel computational framework for optimal holographic stimulation of neural ensembles.

## 3 Methods

### 3.1 Optogenetic receptive field model

Given a population of $N$ neurons, we first wish to learn ORF functions $g_n$ that model how each neuron responds to optogenetic stimulation at different laser powers and locations near the cell soma. Ideally, ORF mapping should be completed quickly, stimulating as few times as possible, so that stimulus optimisation and the desired experiment can begin. Therefore, an ORF model should leverage prior knowledge of how neurons respond to stimulation. Importantly, spike probability should increase monotonically with laser power until saturation [42], and the ORF should approximately match the shape of the somatic membrane (though enlarged to account for the $\sim 10 \ \mu$m diameter of the

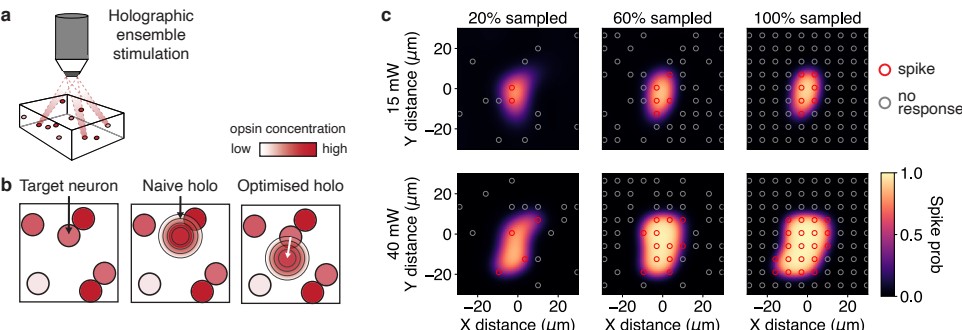

Figure 1: Off-target stimulation with two-photon optogenetics. (a) Two-photon holographic opto-genetics can be used to elicit spikes in specific ensembles of experimenter-selected neurons. (b) OTS arises due to an inability to confine two-photon excitation to the soma of a target neuron. By repositioning the hologram away from the soma, OTS could be avoided while still activating the target neuron. (c) Data from real two-photon optogenetics experiment showing that at high power (e.g. 40 mW) a neuron can be activated from 15-20 $\mu$m away, though this depends on the specific pattern of opsin expression at the soma and proximal dendrites. Red and gray circles indicate locations where stimulation resulted in successful or unsuccessful spikes. Colour map shows the inferred probability of spiking (i.e., the ORF) using the log-barrier Newton method from Subsection 3.2. As the number of sampled locations and laser powers increases, the GP model adapts to the particular ORF shape (see supplementary material for the prior mean). This shape can then be exploited to precisely optimise holographic stimuli. Data from PV-neuron in L2/3 of V1 expressing the soma-targeted, excitatory opsin ChroME2f [44].

holographic disk [22]). However, any residual expression of opsin molecules in the proximal dendrites of a neuron creates unique differences in its ORF shape compared to other neurons (Figure 1c). Further, neurons can express opsin in variable concentrations and vary in their intrinsic excitability, leading to an unpredictable dependence of spiking on power. Thus, we need an approach to modelling an ORF that roughly describes the typical shape and photosensitivity of a cell, but that can flexibly adapt to variation across neurons.

To simultaneously account for these factors, we use a novel variant of the GP-Bernoulli model. Let $y_{nt} \in \{0, 1\}$ denote the response of neuron $n$ on trial $t$ to a holographic stimulus $\mathbf{x}_t \in \mathbb{R}^{J \times 3}$. As our approach is designed for all-optical experiments, the response to stimulation is assumed to be observed for all neurons at once using calcium or voltage imaging. Each stimulus delivers two-photon excitation to $J \geq 1$ different target locations, with $\mathbf{x}_t^j = (c_{1t}^j, c_{2t}^j, I_t^j) \in \mathbb{R}^3$ representing the two-dimensional coordinates and laser power of the $j$th target (though note that we can also handle three-dimensional hologram coordinates without any substantial changes to the model). While in some cases the targets will be far enough apart in space to not interact with each other, frequently a neuron will integrate two-photon excitation from multiple nearby targets, increasing the risk of OTS. To account for this, we model the probability of evoking a spike in neuron $n$ by summing across all $J$ points on that neuron's ORF,

$$y_{nt} \sim \text{Bernoulli}(\sigma(\gamma_n(\mathbf{x}_t) - \theta_n)), \quad \gamma_n(\mathbf{x}_t) = \sum_{j=1}^{J} g_n(\mathbf{x}_t^j), \tag{1}$$

where $\sigma$ is the logistic sigmoid function and $\theta_n \in \mathbb{R}$ is a scalar threshold. We model each ORF using a three-dimensional GP prior that describes the effect of stimulating at a given location and power: $g_n \sim \mathcal{GP}(m_n(\cdot), k(\cdot, \cdot))$, where $m_n$ and $k$ are respectively the mean and covariance functions of the GP. During inference (as discussed below) we constrain $g_n$ to be non-negative, which ensures that stimulation cannot have an inhibitory effect. Further, using a non-negativity constraint rather than (e.g.) applying a rectifying nonlinearity to $g_n$ guarantees that the posterior distribution remains log-concave, facilitating tractable identification of the global optimum.

To encode prior knowledge about optogenetic stimulation, the mean function should have the property that the probability of evoking a spike increases with laser power but decays quickly as the hologram

is positioned further away from the nucleus. To this end, we set

$$m_n(\mathbf{x}) = \rho I \exp(-\|\mathbf{c} - L_n\|^2/\sigma_m^2) \tag{2}$$

for $n = 1, \ldots, N$ (note that we suppress reference to the target dimension $j$ for notational clarity here and in the definition of the covariance function below). Here $\mathbf{c} = (c_1, c_2)$ denotes the coordinates of the holographic target, $L_n$ is the location of neuron $n$, and $\rho$ can be loosely interpreted as the average opsin expression level, determining how laser power affects the probability of spiking. While we treat $\rho$ and $\sigma_m^2$ as static hyperparameters, in future work one could consider a hierarchical model where the mean function is also learned, with its accuracy improving as additional ORFs are probed.

We model the covariance between points on the ORF using the radial basis function (RBF) kernel $k$, which regularises the ORF by encouraging it to vary smoothly through space and power. However, we note that other covariance kernels could be used here provided that they are differentiable with respect to $\mathbf{x}$. We define

$$k(\mathbf{x}, \mathbf{x}') = \alpha^2 \exp\left(-\frac{1}{2}(\mathbf{x} - \mathbf{x}')^\top \mathbf{\Lambda}(\mathbf{x} - \mathbf{x}')\right) + \sigma_d^2 \delta_{\mathbf{x}, \mathbf{x}'}, \tag{3}$$

where $\mathbf{\Lambda} = \mathrm{diag}(\lambda_s, \lambda_s, \lambda_I)$ is a diagonal matrix with each diagonal term giving the characteristic lengthscale for the corresponding GP dimension. We assume that the lengthscales $\lambda_s$ in the two spatial dimensions are equal (a typical assumption in holography, see e.g. ref. [6]), but take a different lengthscale $\lambda_I$ for the laser power dimension. Finally, we denote the hyperparameters of the ORF prior by $\phi = (\alpha^2, \sigma_d^2, \lambda_s, \lambda_I, \rho, \sigma_m^2)$.

## 3.2 Inference

During an ORF mapping experiment we first collect calibration data by performing stimulation in a series of trials $\mathbf{x}_1, \ldots, \mathbf{x}_T$, where each $\mathbf{x}_t$ targets $J$ different locations and laser powers near a random set of neurons. Then we combine these stimuli with the corresponding observations of any evoked activity $\mathbf{y}_{:,1}, \ldots, \mathbf{y}_{:,T}$ to estimate the ORFs. Here each $\mathbf{y}_{:,t} \in \{0, 1\}^N$ describes the population response to the stimulus $\mathbf{x}_t$. Due to the non-negativity constraints on $g_n$, the posterior $p(g_n \mid \mathbf{y}_n, \{\mathbf{x}_t\}_{t=1}^T, \theta_n, \phi)$ is not itself a GP, and hence we perform maximum *a posteriori* (MAP) inference rather than attempting to obtain a complete description of posterior uncertainty (though see the supplementary material for a full treatment of posterior uncertainty in the $J = 1$ case). To this end, the MAP estimates are obtained by maximising the log-posterior, which, via Bayes rule, is equivalent to computing

$$\hat{g}_n, \hat{\theta}_n = \underset{g_n, \theta_n}{\mathrm{argmax}} \left\{ \sum_{t=1}^T \ln p(y_{nt} \mid \mathbf{x}_t, g_n, \theta_n) + \ln p(g_n(\mathbf{x}_1), \ldots, g_n(\mathbf{x}_T) \mid \phi) \right\} \tag{4}$$

$$\text{such that } g_n(\mathbf{x}_t) \geq 0 \text{ for } t = 1, \ldots, T.$$

Note that we have abbreviated $\hat{g}_n(\mathbf{X})$ (the MAP version of the ORF) as $\hat{g}_n$. To solve Equation 4 we alternate between using Newton's method with a log-barrier to update $g_n$ (such that the optimised ORFs respect the non-negativity constraints), and performing gradient descent steps for updating $\theta_n$. We also adaptively set the Newton step-size using a standard backtracking line-search method.

## 3.3 Optimisation approach

Given the estimated ORFs, we then aim to identify holographic stimuli that evoke specific neural activity patterns as accurately as possible (i.e., minimising off-target activation). To do this, we introduce a novel optimisation approach that exploits properties of the GP to compute gradients of an error function between a target activity pattern and the (predicted) evoked activity. To this end, let $\mathcal{G} = \{\hat{g}_n, \hat{\theta}_n\}_{n=1}^N$ be the MAP-estimated ORFs, and define the predicted evoked activity as

$$\hat{y}(\mathbf{x}, \mathcal{G}) = (\sigma(\hat{\gamma}_1(\mathbf{x}) - \hat{\theta}_1), \ldots, \sigma(\hat{\gamma}_N(\mathbf{x}) - \hat{\theta}_N)) \in \mathbb{R}^N, \tag{5}$$

where $\hat{\gamma}_n(\mathbf{x}) = \sum_{j=1}^J \hat{g}_n(\mathbf{x}^j)$. Let $\mathbf{\Omega} \in \{0, 1\}^N$ denote a target activity pattern. The optimal holographic stimulus is then

$$\mathbf{x}_{\mathrm{optimal}} = \underset{\mathbf{x}}{\mathrm{argmin}} \|\mathbf{\Omega} - \hat{y}(\mathbf{x}, \mathcal{G})\|^2 \text{ such that } 0 \leq I \leq I_{\mathrm{max}}. \tag{6}$$

Note that we include $I_{\max}$ as an upper bound on the laser power, which can be set to match the power practically deliverable by the microscopy system or to prevent tissue damage due to excess heating.

We propose to solve Equation 6 using a projected gradient descent algorithm. While for the ORF mapping phase we only need to compute the ORFs $\hat{g}_n$ at a small set of stimulation points $\mathbf{x}_t$, for the optimisation phase we must evaluate the gradients of Equation 6 (and therefore of the ORFs) at a series of new, unobserved points constituting an optimisation path. However, it is not immediately obvious how to evaluate these gradients for a general nonparametric function $g_n$. The key idea for our approach is that at each step, gradient descent will provide an updated estimate $\mathbf{x}^*$ of the optimal stimulus, which will act as a "test point" commonly used in GP regression. We then leverage properties of the GP to perform *inference* of the ORF gradients at $\mathbf{x}^*$, and subsequently perform a gradient descent update that reduces the value of the objective function in Equation 6.

Inferring the ORF gradients relies on the fact that a GP and its derivative are jointly GP-distributed, and hence the derivative can be inferred just from a small number of observations of the GP itself. To this end, note that the covariance between a GP and its derivative is given by [45, Sec 9.4]

$$\mathrm{cov}\left(g_n(\mathbf{x}_t), \frac{\partial}{\partial x_d^*}g_n(\mathbf{x}^*)\right) = \frac{\partial k(\mathbf{x}_t, \mathbf{x}^*)}{\partial x_d^*}, \tag{7}$$

where $d$ is any dimension of $\mathbf{x}^*$ and where we have suppressed reference to the hologram target $j$ for notational clarity here and in the following equation. The expression in Equation 7 can be used to obtain a predictive distribution over derivative functions consistent with the observed data points, which we use to evaluate the derivatives of $\hat{g}_n$ at the test point $\mathbf{x}^*$ via

$$\frac{\partial \hat{g}_n(\mathbf{x}^*)}{\partial x_d^*} = \frac{\partial m_n(\mathbf{x}^*)}{\partial x_d^*} + \mathrm{cov}\left(g_n(\boldsymbol{\mathcal{X}}), \frac{\partial g_n(\mathbf{x}^*)}{\partial x_d^*}\right)^{\top}\mathbf{K}^{-1}(\hat{g}_n(\boldsymbol{\mathcal{X}}) - m_n(\boldsymbol{\mathcal{X}})), \tag{8}$$

where $\boldsymbol{\mathcal{X}} \in \mathbb{R}^{JT \times 3}$ is the collection of $JT$ unique points on the ORF that were probed during ORF mapping, $g_n(\boldsymbol{\mathcal{X}}), m_n(\boldsymbol{\mathcal{X}}) \in \mathbb{R}^{JT}$ give the value of the GP and mean function evaluated at each such point, and where $\mathbf{K}$ is the corresponding GP covariance matrix obtained by evaluating the covariance kernel $k$ at every pair of elements of $\boldsymbol{\mathcal{X}}$. Further details are provided in the supplementary material.

Equation 8 allows us to define a closed-form gradient of the objective function, which we use in a projected gradient descent algorithm (Algorithm 1). Note, however, that the objective function is not convex in $\mathbf{x}$, and therefore such updates are only guaranteed to converge to a locally optimal solution. We therefore typically run Algorithm 1 with several random initialisations and select the optimised target with the lowest predicted error.

---

**Algorithm 1: Bayesian target optimisation (Bataro).**

---

1 Compute MAP estimates of ORFs $\{\hat{g}_n, \hat{\theta}_n\}_{n=1}^N$ from calibration data $\{\mathbf{y}_n\}_{n=1}^N, \{\mathbf{x}_t\}_{t=1}^T$ using Newton's method with log-barrier.
2 Initialise targets $\mathbf{x} \in \mathbb{R}^{J \times 3}$ to random locations near the somas of the $J$ target neurons and with random laser powers.
3 **while** *target not converged* **do**
4 $\quad$ Construct gradient vectors $\nabla_{\mathbf{x}}\hat{\gamma}_n(\mathbf{x})$ for $n = 1, \dots, N$ using inference of ORF derivatives (Equation 8).
5 $\quad$ Set $\boldsymbol{\delta}_{\mathbf{x}} = -2\sum_{n=1}^N(\Omega_n - \sigma(\hat{\gamma}_n(\mathbf{x}) - \hat{\theta}_n))\sigma'(\hat{\gamma}_n(\mathbf{x}) - \hat{\theta}_n)\nabla_{\mathbf{x}}\hat{\gamma}_n(\mathbf{x})$.
6 $\quad$ Perform gradient descent update $\mathbf{x} \leftarrow \mathbf{x} + \beta\boldsymbol{\delta}_{\mathbf{x}}$ with step-size $\beta$.
7 $\quad$ Project laser power onto feasible domain, $I_j \leftarrow \min(I_j, I_{\max})$ for $j = 1, \dots, J$.
8 **end**

---

## 4 Results

### 4.1 Simulations

We first tested Bataro by simulating an optical "write-in" experiment, where we attempted to write specific activity patterns into a hypothetical neural population. To do this, we positioned 50 neurons randomly in a 250 $\mu$m $\times$ 250 $\mu$m field of view (FOV), sampled each neuron's ORF from its GP

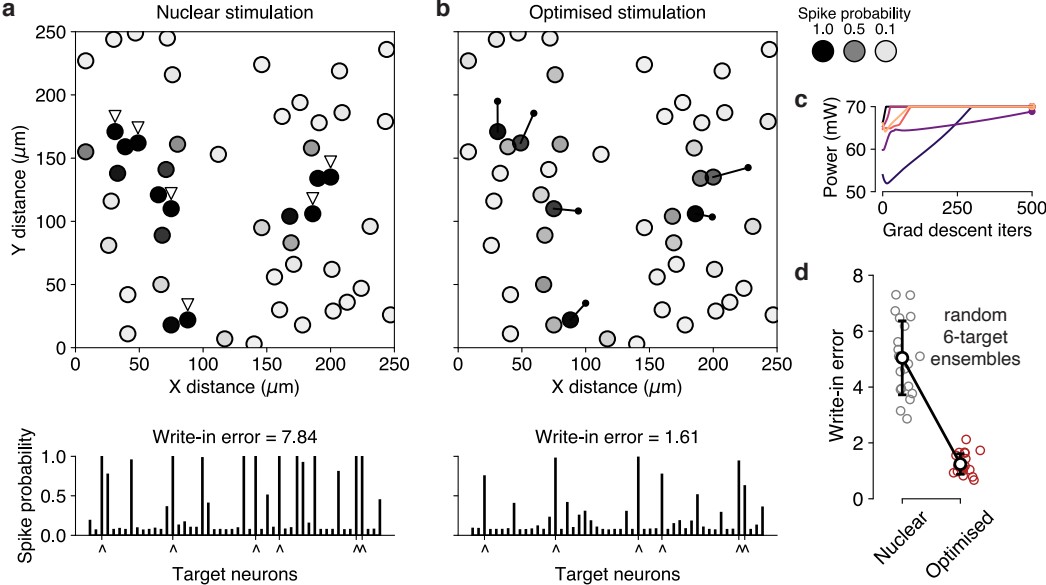

Figure 2: Minimising off-target stimulation using Bayesian target optimisation. (a) Direct nuclear stimulation at 70 mW successfully activates the target neurons with high probability, but also activates nearby non-target neurons due to OTS. Triangles indicate target neurons. Shading indicates probability of spiking. Optical write-in error (bottom) given as the sum of squared errors between the evoked and desired activity patterns. (b) Optimised stimulation using Algorithm 1 repositions the hologram locations away from the nuclei of non-target neurons, resulting in a substantial reduction in off-target activation. (c) Optimisation trajectory of the 6 different target laser powers. Initial laser powers selected randomly between 50 and 70 mW. (d) Optimising holographic targets for 20 different random ensembles shows a robust reduction of optical write-in error (average reduction, 75%).

prior, and generated responses to ensemble stimulation. To quantify the optical write-in error, we used the sum of squared errors $\sum_n (\Omega_n - y_n(\mathbf{x}))^2$, where $y_n(\mathbf{x})$ is the ground truth probability of neuron $n$ spiking in response to stimulus $\mathbf{x}$. This error takes the interpretable value of $\sim m$ if $m$ non-target neurons are inadvertently recruited due to OTS. In this simulation, holographic stimulation of an example ensemble consisting of 6 selected neurons resulted in a substantial recruitment of activity from non-target neurons due to OTS, especially when multiple holograms converged on nearby non-target neurons (Figure 2a; optical write-in error = 7.84).

To optimally reposition the holographic targets, we probed the population's ORFs using 10-target ensemble stimulation, and then used Bataro (Algorithm 1). This successfully repositioned holographic targets an appropriate distance from the nuclei of the target neurons, such that while minimal excitation was applied to the non-target neurons, the target neurons were still driven to spike with high probability (Figure 2b,c; write-in error = 1.61). To confirm that the improvement in write-in precision was robust, we repeated the optimisation for 20 additional random 6-target ensembles and found that off-target effects were substantially reduced in every case, with the write-in error reducing by 75% on average compared to naively stimulating the nucleus of each neuron (Figure 2d).

Next, we investigated two variables known to critically constrain the ability to evoke specific neural activity patterns: (1) the density of opsin-expressing neurons in the stimulation FOV, and (2) the size of the ensemble that one is attempting to activate. Figure 3a illustrates how increasing the number of opsin-expressing neurons introduces many more potential non-target neurons that could be inadvertently activated. This causes errors induced by direct nuclear stimulation of (e.g.) 5-neuron ensembles to grow rapidly (Figure 3b, grey curve), though note that the magnitude of the error depends on many experimental factors including the size of the ORF, numerical aperture of the microscope, and the soma-targeting efficacy. By optimising holographic target locations and laser powers, the average write-in error was reduced by 76% relative to nuclear stimulation (Figure 3b, red curve). Similarly, in these simulations, nuclear stimulation resulted in an error that grew almost

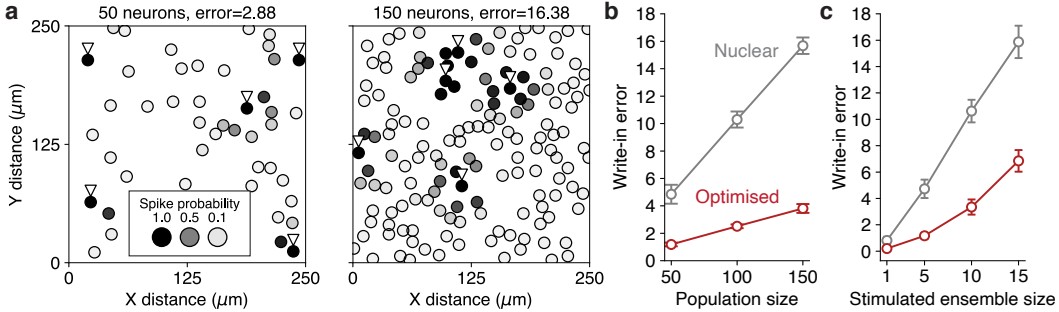

Figure 3: Performance of Bayesian target optimisation in increasingly difficult contexts. (a) Two example scenarios with low (left) and high (right) density of opsin-expressing neurons. Triangles indicate example 5-target ensemble to be stimulated. At low density, the risk of OTS is low because neurons are often spaced far apart. However, at high density, OTS arising from direct nuclear stimulation at high power is unavoidable. (b) Optical write-in error for nuclear and optimised stimulation of 5-target ensembles. Error bars show the mean error $\pm$ 1 s.d. over 10 different simulations. For each simulation we averaged the write-in error over 20 random ensembles. (c) Same as (b), but for a fixed population size of 50 and varying ensemble size.

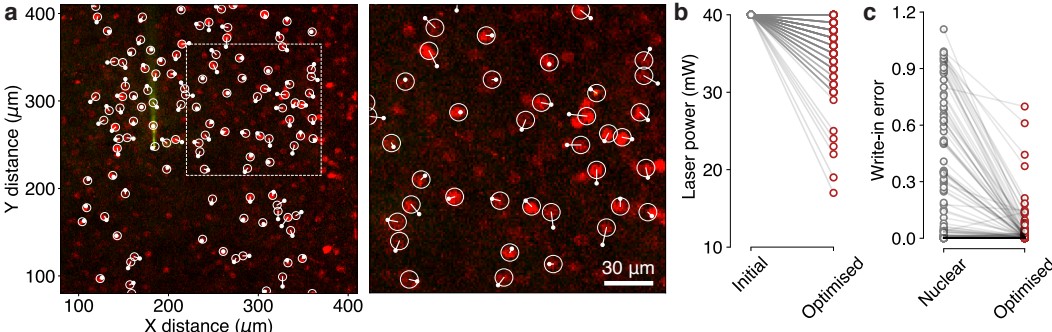

Figure 4: Performance of Bayesian target optimisation using simulations based on two-photon holographic optogenetics data. (a) Left: field of view from a real *in vitro* optogenetics experiment showing every optimised single-target hologram. Opsin fused to red fluorescent protein mRuby3 so that opsin-expressing neurons can be visualised. Unfilled white circles represent putative neurons detected by automated cell segmentation method. Smaller filled white circles represent optimised holographic targets, shown in relation to the cell nucleus by a straight white line. Right: zoomed view of optimised targets, corresponding to dashed region in the left panel. (b) Optimised laser powers relative to their initialised values show how Bataro exploits differences in photosensitivity to avoid OTS. Each circle represents the power delivered to a single holographic target. (c) Reduction of optical write-in error using optimised holographic targets. Each circle represents the error when attempting to stimulate a single neuron.

directly proportionally to the size of the stimulated ensemble, indicating that almost as many non-target neurons were activated as target neurons. However, Bataro reduced the average write-in error by 69% across ensembles of size 1-15 (Figure 3c).

## 4.2 Application to holographic optogenetics experiments

To test whether Bataro could eliminate OTS in realistic settings, we created synthetic optogenetics experiments involving >100 neurons from a small number ($n$=4) of detailed cell-attached recordings in slice [42]. Briefly, each slice experiment was performed by first establishing a cell-attached electrophysiological recording of a single L2/3 interneuron from mouse V1 expressing the soma-targeted opsin ChroME2f. Then, the recorded neuron was optogenetically stimulated at a dense grid

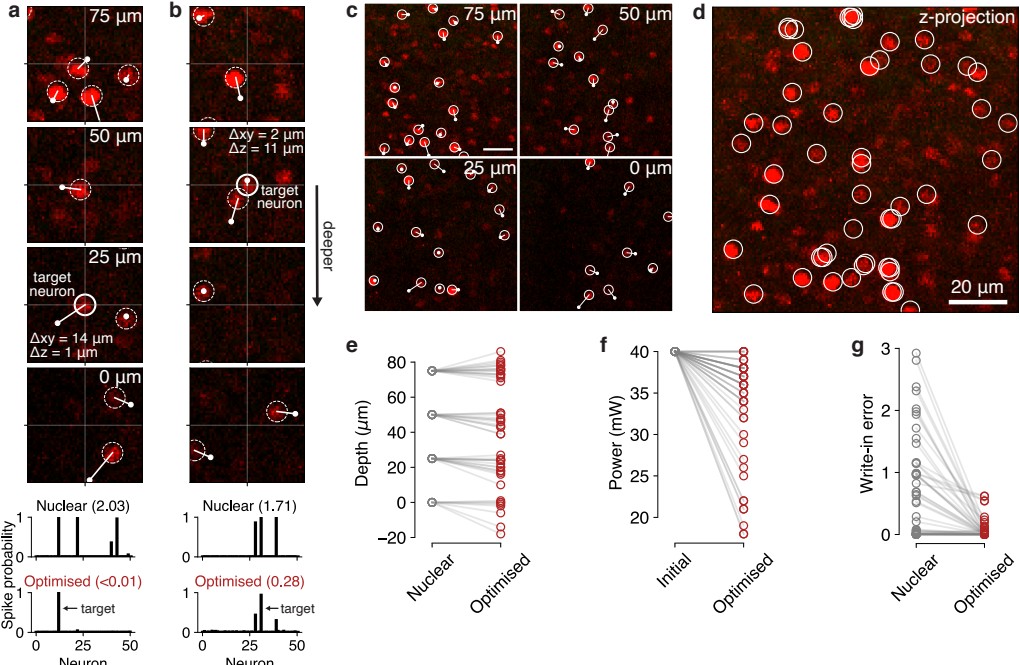

Figure 5: Optimisation of holographic targets in three-dimensional space. (a) Example target neuron and optimised holographic stimulus (plane 3, 25 $\mu$m). By repositioning the hologram (primarily in the x/y dimensions), off-target activation is entirely eliminated (bottom spike probability plots, inset numbers show write-in error). Deepest plane labelled as 0 $\mu$m by convention. Solid white circle indicates target neuron. Dashed white circles indicate nearby non-target neurons that must be avoided. (b) Similar to (a), but for a neuron in plane 2 (50 $\mu$m) and with repositioning of the hologram primarily in the z dimension. (c) All optimised single-target holograms over four stimulation planes. Targets shown at their original depth for visualisation. Average displacement of optimised holographic targets relative to nuclei, 8.3 $\mu$m. Scale bar, 20 $\mu$m. (d) Max-projection over z-planes simultaneously showing locations of all segmented opsin-expressing neurons. (e) Optimised depths of holographic targets corresponding to neurons in (c). Depths are shown in comparison to one of four depths that the target neuron was segmented at during the experiment. (f) Optimised laser powers relative to their initialised values. (g) Reduction of optical write-in error for all neurons, across multiple depths.

of locations surrounding the cell and at a range of laser powers to comprehensively map its ORF (see Figure 1c, "100% sampled" column, also see supplementary material for additional examples). We used this data to create a set of four "ground truth" ORFs by fitting the GP-Bernoulli model from Equation 1, where the hyperparameters of the GP covariance kernel were selected using the cross-validated predictive log-likelihood. Next, we used a fluorescence image from a separate experiment to extract the locations of 116 putative opsin-expressing neurons, and randomly assigned each putative neuron one of four ground-truth ORFs. Together, this enabled us to simulate responses to optogenetic stimulation at arbitrary locations and laser powers and for a large number of neurons that were realistically distributed in space.

We sampled responses to optogenetic stimulation at a sparse grid of locations surrounding each neuron to map their ORFs in a hypothetical experiment (see supplementary methods for details). Then, we used Bataro to compute single-target stimuli that activated each neuron individually and minimised OTS (Figure 4a). We found that, on the one hand, a subset of neurons required no change from nuclear stimulation at high power due to being spatially isolated. However, regions with large numbers of closely-packed, opsin-expressing neurons required repositioning of the holographic targets (Figure 4a, right) and adjustment to the laser powers (Figure 4b) to eliminate OTS (average distance between nuclear and optimised stimuli, 4.9 $\mu$m). Across all single-target holograms, target optimisation reduced the average write-in error by 85% (Figure 4c). We also performed a control

analysis by matching the nuclear stimulation laser power to the average optimised laser power and obtained similar results (Figure S3).

Some holographic microscopy systems cannot position holograms at arbitrary continuous depths, instead either requiring holograms to be positioned in two dimensions or on a preselected number of discrete stimulation planes. Hence, thus far, we focused our efforts on the much more common case of two-dimensional optogenetics experiments. However, OTS is generally observed to be even more prevalent in three dimensions, as the optical point spread function is elongated in the "z"-axis [15, 6, 13, 17]. We therefore tested Bataro in the more general setting of three spatial dimensions and laser power (i.e., four dimensions in total), under the assumption that some microscopy systems may be able to flexibly reposition holograms continuously across depths.

We used the fact that the cell-attached recordings were repeated at multiple depths to generate synthetic optogenetics experiments with four-dimensional ORFs (Figure 5a,b). Optimisation of holographic targets successfully reduced OTS by exploiting both the x/y dimensions (Figure 5a), as well the z dimension (Figure 5b), in addition to automatically adjusting the laser power where needed. Repeating the optimisation for every single-target hologram reduced the average optical write-in error by 89% (from 0.7 to 0.08, Figure 5c-g; average distance between nuclear and optimised targets, 8.3 $\mu$m), demonstrating the feasibility of largely eliminating OTS in optogenetics experiments involving all three spatial dimensions.

## 5 Conclusions

We developed Bataro, a novel computational framework for two-photon optogenetics that optimises holographic stimuli to evoke neural activity patterns with minimal off-target activation. Our preliminary *in vitro* experiments predict that specific adjustments to holographic target placements of 5-10 $\mu$m (on average) could substantially reduce OTS, even in the more challenging regime of three-dimensional optogenetic stimulation. Our key idea was to simultaneously exploit variability in photoexcitability, residual opsin expression in proximal dendrites (due to imperfections in opsin soma-targeting), and the specific spatial arrangement of neurons in the stimulation FOV. Optimisation using these three factors ultimately led to reductions in the optical write-in error of 85-89% for single-target holograms using data from *in vitro* two-photon optogenetics experiments.

Whether achieving such stimulus precision is worth spending valuable experiment time mapping ORFs is, of course, highly dependent on the application. While for some experiments minimising off-target activation may not be critical, there are many cases where stimulating with as close to "true" single-cell precision as possible is essential to testing a scientific hypothesis. For example, "hub" neurons in the hippocampus are believed to orchestrate network-level function through their unilateral activity [46, 47]. Thus, using two-photon optogenetics to test whether an individual neuron represents a functional hub by stimulating that neuron alone requires true single-cell precision. In such cases, we believe devoting experiment time to ORF mapping and stimulus optimisation is a necessary trade-off.

Throughout our analysis we have assumed that ORFs are stable over time (i.e., are stationary). However, desensitisation of opsins following prolonged illumination has previously been observed [11], indicating that some nonstationarity could be encountered in real experiments. To account for this, we note that after the ORFs have been mapped and holographic stimuli optimised, it is straightforward to periodically recompute the MAP estimate of the ORF as the experiment proceeds. This would provide an updated estimate of the laser power required to maintain optimal precision. We hypothesise that this would not require repeating the entire ORF mapping phase as we do not expect the shape of the ORF to change drastically beyond desensitisation.

Standard GP regression methods scale cubically with the number of locations and laser powers probed, and therefore estimation of each neuron's ORF could encroach on experiment time if ORFs are mapped at high detail just due to required computation time. However, computational efficiency could be improved by appealing to modern GP techniques such as inducing point methods [48], or by adaptively mapping ORFs using Bayesian active learning to minimise the number of probed locations and powers [49, 50]. While in this work we have focused primarily on demonstrating the feasibility of overcoming OTS, in future work we plan to minimise both required experiment and computation time in order to deploy Bataro online.

## Acknowledgements

We thank Darcy Peterka, Benjamin Antin, Cole Hurwitz, Shizhe Chen, and Ben Shababo for helpful discussions and suggestions. This work was funded by NIH awards 1RF1MH120680 and 1U19NS107613-01 to HA and LP. MAT and LP were supported by the Gatsby Charitable Foundation and NSF NeuroNex award 1707398.

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

# 6 Supplementary material

## 6.1 Animal ethics statement

All experiments on animals were conducted with approval of the Animal Care and Use Committee of the University of California, Berkeley.

## 6.2 Compute

All computational procedures were performed either on a desktop workstation running Ubuntu 18.04 with an Intel Xeon E5-2620 v4 CPU, four GTX 1080 Ti GPUs, and 112GB RAM, or on the Axon computer cluster based at the Zuckerman Institute (Columbia University) using nodes comprised of two Xeon E5-2660 v4 CPUs, eight GTX 1080 Ti GPUs, and 125GB RAM.

## 6.3 Broader societal impact

Our work is significant for interventional approaches to studying the brain and its connection to disease. By minimising off-target activation, Bayesian target optimisation could enable (e.g.) more precise synaptic connectivity mapping, improving our understanding of neural circuitry. This advancement has potential implications for understanding brain disorders like epilepsy, where abnormal synaptic connections are central to seizure generation and propagation. Deepening our understanding of these diseases can lead to enhanced targeted interventions and more effective therapeutic strategies, benefiting individuals with neurological disorders.

## 6.4 Code availability

An open-source implementation of Bayesian target optimisation is available in Python at `https://github.com/marcustriplett/bataro`.

## 6.5 Single-target holographic stimulus optimisation with posterior uncertainty

Here we provide further mathematical details for optimising holographic stimuli. First, we develop the approach for single optogenetic targets, as this is most closely related to existing GP-based receptive field inference techniques. The single-target case also allows us to have a full treatment of posterior uncertainty (unlike for optimising ensemble stimuli) which may be desired in certain applications.

**Optogenetic receptive field model.** We use a GP-Bernoulli approach to model the response $y_{nt}$ of neuron $n$ on trial $t$ to a single-target stimulus $\mathbf{x}_t$,

$$y_{nt} \sim \text{Bernoulli}(\sigma(g_n(\mathbf{x}_t))), \tag{9}$$

where the stimulus $\mathbf{x}_t = (c_{1t}, c_{2t}, I_t) \in \mathbb{R}^3$ represents the two-dimensional coordinates and laser power of the $t$-th hologram. Each ORF follows a three-dimensional GP prior $g_n \sim \mathcal{GP}(m_n(\cdot), k(\cdot, \cdot))$, where $m_n$ and $k$ again are the mean and covariance functions of the GP.

**Posterior inference.** Unlike for ensemble stimulation, for single-target stimulation we do not require that the ORF $g_n$ is non-negative. This is because now if a point on the ORF becomes inhibitory (by taking a negative value), it will not conflict with excitation from any other hologram. Consequently, the posterior of $g_n$ is a GP, which allows us to work with a full description of posterior uncertainty. To compute the posterior, we use the conventional Laplace approximation. Briefly, this consists of approximating the posterior using a multivariate normal $q(g_n \mid \boldsymbol{\mu}_n, \boldsymbol{\Sigma}_n) = \text{Normal}(g_n \mid \boldsymbol{\mu}_n, \boldsymbol{\Sigma}_n) \approx p(g_n \mid \mathbf{y}_n, \mathbf{X}, \phi)$. The mean $\boldsymbol{\mu}_n$ is obtained by maximising the log-posterior, given by the expression

$$\ln p(g_n \mid \mathbf{y}_n, \mathbf{X}, \phi) = \sum_{t=1}^{T} \ln p(\mathbf{y}_{nt} \mid \mathbf{x}_t, g_n) + \ln p(g_n(\mathbf{x}_1), \dots, g_n(\mathbf{x}_T) \mid \phi) + \text{const}, \tag{10}$$

where $\mathbf{X} = (\mathbf{x}_1, \dots, \mathbf{x}_T)$ and where const does not depend on $g_n$. Since the posterior is log-concave in $g_n$, we use Newton's method to identify the global optimum of Equation 10, and adaptively set the Newton step-size using a standard backtracking line-search method. Letting $\mathbf{H} = \nabla\nabla_{g_n} \ln p(g_n \mid$

$\mathbf{y}_n, \mathbf{X}, \phi)$ be the Hessian of the log-posterior, the posterior covariance matrix is obtained by setting $\boldsymbol{\Sigma}_n = -\mathbf{H}^{-1}\mid_{g_n=\mu_n}$.

**Target optimisation**. Let $G = (g_1, \ldots, g_N)$, and define the predicted evoked activity for single holographic targets as $\hat{y}(\mathbf{x}, G) = (\sigma(g_1(\mathbf{x})), \ldots, \sigma(g_N(\mathbf{x})))$. To minimise the error between a target binary activity pattern $\boldsymbol{\Omega} \in \{0, 1\}^N$ and the predicted evoked activity, we solve an optimisation problem that accounts for the uncertainty in the ORF estimates:

$$\mathbf{x}_{\text{optimal}} = \underset{\mathbf{x}}{\arg\min}\ \mathbb{E}_{q(G|\boldsymbol{\mu},\boldsymbol{\Sigma})} \left[\|\boldsymbol{\Omega} - \hat{y}(\mathbf{x}, G)\|^2\right] \quad \text{such that} \quad 0 \le I \le I_{\max}, \tag{11}$$

where $q(G \mid \boldsymbol{\mu}, \boldsymbol{\Sigma}) = \prod_{n=1}^N q(g_n \mid \boldsymbol{\mu}_n, \boldsymbol{\Sigma}_n)$ gives the joint posterior across all ORFs. To solve Equation 11, we first sample ORFs $g_n^{(s)}$ (for $s = 1, \ldots, S$) from their posterior distributions to approximate the expected error at the current estimate $\mathbf{x}^*$,

$$\mathbb{E}_{q(G|\boldsymbol{\mu},\boldsymbol{\Sigma})} \left[\|\boldsymbol{\Omega} - \hat{y}(\mathbf{x}^*, G)\|^2\right] \approx \frac{1}{S} \sum_{s=1}^S \sum_{n=1}^N \left(\Omega_n - \sigma(g_n^{(s)}(\mathbf{x}^*))\right)^2. \tag{12}$$

Then, we compute the partial derivative (in dimension $d$) of the expected error by differentiating through the Monte Carlo approximation,

$$\frac{\partial}{\partial x_d^*} \mathbb{E}_{q(G|\boldsymbol{\mu},\boldsymbol{\Sigma})} \left[\|\boldsymbol{\Omega} - \hat{y}(\mathbf{x}^*, G)\|^2\right] \approx -\frac{2}{S} \sum_{s=1}^S \sum_{n=1}^N (\Omega_n - \sigma(g_n^{(s)}(\mathbf{x}^*)))\sigma'(g_n^{(s)}(\mathbf{x}^*))\frac{\partial}{\partial x_d^*} g_n^{(s)}(\mathbf{x}^*). \tag{13}$$

Next we must evaluate the partial derivative on the right-hand side of Equation 13. We use the fact that a GP and its derivative are jointly GP-distributed, and hence infer the derivative from observations of the ORF. The covariance between a GP and its derivative is given by [45, Sec 9.4]

$$\text{cov}\left(g_n(\mathbf{x}_t), \frac{\partial}{\partial x_d^*} g_n(\mathbf{x}^*)\right) = \frac{\partial k(\mathbf{x}_t, \mathbf{x}^*)}{\partial x_d^*} = \frac{\alpha^2}{\lambda_d^2}(x_{dt} - x_d^*)\exp\left(-\frac{\|\mathbf{x}_t - \mathbf{x}^*\|^2}{2\lambda_d^2}\right), \tag{14}$$

where the second equality is specific to the RBF covariance. Thus, we can use Equation 14 to obtain the posterior predictive mean for the derivative GPs in closed form as [51, Sec 2.7]

$$\mathbb{E}_{q(g_n|\boldsymbol{\mu}_n,\boldsymbol{\Sigma}_n)}\left[\frac{\partial g_n(\mathbf{x}^*)}{\partial x_d^*}\right] = \frac{\partial m_n(\mathbf{x}^*)}{\partial x_d^*} + \text{cov}\left(g_n(\mathbf{X}), \frac{\partial g_n(\mathbf{x}^*)}{\partial x_d^*}\right)^\top \mathbf{K}^{-1}(\boldsymbol{\mu}_n - m_n(\mathbf{X})). \tag{15}$$

Here $\mathbf{X} = (\mathbf{x}_1, \ldots, \mathbf{x}_T)$ is the collection of unique points on the ORF probed during calibration. If Equation 15 is combined with an expression for the posterior predictive variance, one obtains a full predictive distribution over derivative functions consistent with the observed neural responses. However, rather than working with this full distribution, we instead use Equation 15 to approximate the derivatives of the Monte Carlo samples by replacing the posterior mean $\boldsymbol{\mu}_n$ with a Monte Carlo sample,

$$\frac{\partial g_n^{(s)}(\mathbf{x}^*)}{\partial x_d^*} \approx \frac{\partial m_n(\mathbf{x}^*)}{\partial x_d^*} + \text{cov}\left(g_n(\mathbf{X}), \frac{\partial g_n(\mathbf{x}^*)}{\partial x_d^*}\right)^\top \mathbf{K}^{-1}(g_n^{(s)}(\mathbf{X}) - m_n(\mathbf{X})). \tag{16}$$

Equation 16 then allows us to define a closed-form approximate gradient $\tilde{\nabla}_{\mathbf{x}^*} g_n^{(s)}$ at test point $\mathbf{x}^*$, defined as

$$\tilde{\nabla}_{\mathbf{x}^*} g_n^{(s)} = \left[\frac{\partial g_n^{(s)}(\mathbf{x}^*)}{\partial x_1^*}, \ldots, \frac{\partial g_n^{(s)}(\mathbf{x}^*)}{\partial x_D^*}\right]^\top, \tag{17}$$

which we use in the single-target projected gradient descent algorithm (Algorithm 2). Note that one could also consider a quadrature approach to solving Equation 12, which may be more efficient than Monte Carlo sampling. However, the presentation of the Monte Carlo approach is instructive for deriving the optimisation of ensemble stimuli below.

**Algorithm 2:** Projected Monte Carlo gradient descent algorithm for optimising single-target holograms

---

1  Infer ORF posterior $q(G \mid \boldsymbol{\mu}, \boldsymbol{\Sigma})$ from calibration data $\{\mathbf{y}_n\}_{n=1}^N$, $\mathbf{X}$ using the Laplace approximation.
2  Precompute the negative of the Hessian $\mathbf{W}_n = -\nabla\nabla \ln p(\mathbf{y}_n \mid \mathbf{X}, g_n)\mid_{g_n = \boldsymbol{\mu}_n}$ for each $n$.
3  Initialise $\mathbf{x}$ to random location near soma of target neuron and with random laser power.
4  **while** *target not converged* **do**
5      **for** $n = 1, \ldots, N$ **do**
6          Compute mean and variance of posterior predictive distribution at current target estimate $\mathbf{x}$ via $\mu_n(\mathbf{x}) = m_n(\mathbf{x}) + k(\mathbf{X}, \mathbf{x})^\top \mathbf{K}^{-1}(\boldsymbol{\mu}_n - m_n(\mathbf{X}))$, and $\sigma_n^2(\mathbf{x}) = k(\mathbf{x}, \mathbf{x}) - k(\mathbf{X}, \mathbf{x})^\top (\mathbf{K} + \mathbf{W}_n^{-1})^{-1} k(\mathbf{X}, \mathbf{x})$.
7          Sample ORFs at the current target estimate, $g_n^{(s)}(\mathbf{x}) \sim \text{Normal}(\mu_n(\mathbf{x}), \sigma_n^2(\mathbf{x}))$ for $s = 1, \ldots, S$.
8          Construct approximate gradients $\tilde{\nabla}_{\mathbf{x}} g_n^{(s)}$ for $s = 1, \ldots, S$ using Equation 17.
9      **end**
10      Set $\boldsymbol{\delta}_{\mathbf{x}} = -\frac{2}{S}\sum_{s=1}^S \sum_{n=1}^N (\Omega_n - \sigma(g_n^{(s)}(\mathbf{x})))\sigma'(g_n^{(s)}(\mathbf{x}))\tilde{\nabla}_{\mathbf{x}} g_n^{(s)}(\mathbf{x})$ as per Equation 13.
11      Perform gradient descent update, $\mathbf{x} \leftarrow \mathbf{x} + \beta\boldsymbol{\delta}_{\mathbf{x}}$ with step-size $\beta$.
12      Project laser power onto feasible domain, $I \leftarrow \min(I, I_{\max})$.
13  **end**

---

### 6.6  Additional details on ensemble stimulus optimisation approach

The approach for optimising holographic ensemble stimuli is based on the approach for single-target optimisation, but modified to account for differences in the ORF model and inference. In particular, we again seek to minimise the error between a target activity pattern $\boldsymbol{\Omega}$ and the predicted evoked activity, but now using the MAP estimates $\mathcal{G} = \{\hat{g}_n, \hat{\theta}_n\}_{n=1}^N$ in place of the full posterior distributions. Let $\hat{y}(\mathbf{x}, \mathcal{G}) = (\sigma(\hat{\gamma}_1(\mathbf{x}) - \hat{\theta}_1), \ldots, \sigma(\hat{\gamma}_N(\mathbf{x}) - \hat{\theta}_N))$ be the predicted population response to an ensemble stimulus, where $\hat{\gamma}_n(\mathbf{x}) = \sum_{j=1}^J \hat{g}_n(\mathbf{x}^j)$. The optimal ensemble stimulus is now

$$\mathbf{x}_{\text{optimal}} = \operatorname*{argmin}_{\mathbf{x}} \|\boldsymbol{\Omega} - \hat{y}(\mathbf{x}, \mathcal{G})\|^2 = \operatorname*{argmin}_{\mathbf{x}} \sum_{n=1}^N \left(\Omega_n - \sigma(\hat{\gamma}_n(\mathbf{x}) - \hat{\theta}_n)\right)^2 \tag{18}$$

such that $0 \le I \le I_{\max}$. Evaluating the partial derivative of Equation 18 with respect to dimension $d$ of a test point $\mathbf{x}^*$ yields,

$$\frac{\partial}{\partial x_d^*}\|\boldsymbol{\Omega} - \hat{y}(\mathbf{x}^*, \mathcal{G})\|^2 = -2\sum_{n=1}^N (\Omega_n - \sigma(\hat{\gamma}_n(\mathbf{x}^*) - \hat{\theta}_n))\sigma'(\hat{\gamma}_n(\mathbf{x}^*) - \hat{\theta}_n)\frac{\partial}{\partial x_d^*}\hat{\gamma}_n(\mathbf{x}^*). \tag{19}$$

The derivative on the right-hand side of Equation 19 is given by $\frac{\partial}{\partial x_d^*}\hat{\gamma}_n(\mathbf{x}) = \sum_{j=1}^J \frac{\partial}{\partial x_d^*}\hat{g}_n(\mathbf{x}^j)$, which requires computing the derivative of $\hat{g}_n(\mathbf{x}^j)$. To evaluate this derivative, we use a similar trick to Equation 16, but substituting the MAP estimate in place of the posterior mean or Monte Carlo sample,

$$\frac{\partial}{\partial x_d^*}\hat{g}_n(\mathbf{x}^*) = \frac{\partial}{\partial x_d^*}m_n(\mathbf{x}^*) + \text{cov}\left(g_n(\mathbf{X}), \frac{\partial}{\partial x_d^*}g_n(\mathbf{x}^*)\right)^\top \mathbf{K}^{-1}(\hat{g}_n(\mathbf{X}) - m_n(\mathbf{X})). \tag{20}$$

This expression can also be arrived at by first evaluating the posterior predictive mean of $g_n(\mathbf{x}^*)$, and then differentiating with respect to $x_d^*$.

We use Equation 20 to define a closed-form gradient $\nabla_{\mathbf{x}^*}\hat{\gamma}_n$ at test point $\mathbf{x}^*$ via

$$\nabla_{\mathbf{x}^*}\hat{\gamma}_n = \left[\frac{\partial\hat{\gamma}_n(\mathbf{x}^*)}{\partial x_1^*}, \ldots, \frac{\partial\hat{\gamma}_n(\mathbf{x}^*)}{\partial x_D^*}\right]^\top. \tag{21}$$

Finally, Equation 21 is used in the projected gradient descent algorithm for optimising ensemble stimuli (Algorithm 1).

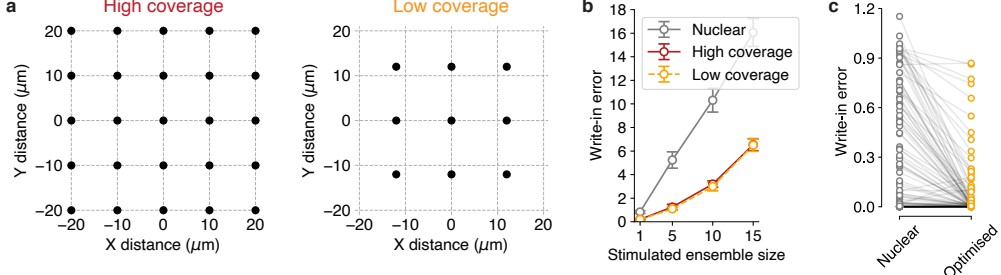

Figure S1: Effect of reducing the number of points at which each ORF is probed. (a) In the high coverage case (left), each ORF is probed by stimulating at a 5×5 grid of points near the soma (grid points separated by 10 $\mu$m), at three different laser powers. In the low coverage case (right), this reduces to stimulating at just a 3×3 grid (grid points separated by 12 $\mu$m) at three powers. However, as the density of opsin-expressing neurons increases, ORFs are probed at high density even in the low coverage case as the grids from different neurons increasingly overlap. (b) Minimal performance difference between the high and low coverage cases in simulations with 50 neurons. (c) Reduction in optical write-in error using cell-attached recordings as in Figure 4, but with low coverage. Reduction in average write-in error, 74% (c.f. 85% with high coverage, Figure 4c).

### 6.7 Further details on simulations and "synthetic" optogenetics experiments

Simulations consisted of both ORF mapping and stimulus optimisation phases. ORF mapping required probing responses to stimulation at a range of laser powers and stimulus locations. We defined a grid of stimulation points surrounding each neuron. In the spatial dimensions, the grid ranged from $-20$ $\mu$m to 20 $\mu$m relative to the centroid of the neuron in steps of 10 $\mu$m, and powers ranged from 30 mW to 70 mW in steps of 20 mW. The complete grid was thus given by the Cartesian product $\{-20, -10, 0, 10, 20\} \times \{-20, -10, 0, 10, 20\} \times \{30, 50, 70\}$. For opsin-expressing neurons that were spaced far apart, this coarse-resolution grid was sufficient because risk of OTS was low, and therefore ORF mapping was not needed at high detail. On the other hand, as the density of opsin-expressing neurons increased, the grids surrounding each neuron increasingly overlapped with each other, resulting in much denser sampling of the ORFs.

For the synthetic optogenetics experiments (based on the cell-attached recordings), we used the same spatial grid spacing but used laser powers of 10, 25, and 40 mW to match the range of powers used in the underlying slice experiment, though note that the slice experiment had a denser spacing than our chosen 15 mW (see example loose-patch recordings below), which we chose to reduce the ORF mapping time. For the optogenetics experiments involving three spatial dimensions, we extended the grid sampling to include depths of $-60$ $\mu$m to 60 $\mu$m in steps of 30 $\mu$m. We also explored the effect of reducing the number of probed grid points to further reduce the time spent mapping ORFs, and found that Bayesian target optimisation maintained high performance when probing with a $3 \times 3$ spatial grid of $\{-12, 0, 12\} \times \{-12, 0, 12\}$ (Figure S1).

We selected the parameters of the GP covariance kernel using 5-fold cross-validation on a separate set of recordings that were made on the same set of four cells, ensuring the hyperparameter selection was using out-of-sample data. Cross-validation was performed using a grid search over a set of possible hyperparameters: the possible radial lengthscales were 2, 4, 8, 16, the power lengthscales were 2, 4, 8, 16, and the amplitudes were 1, 2, 4, 8, 16. For each hyperparameter combination $\theta$ and for each cell, we used Newton's method to fit the GP-Bernoulli model to 80% of the trials in the loose-patch data, yielding an ORF estimate $\hat{g}_\theta$. On the remaining 20% of the trials (denoted as $\mathcal{T}_{\text{held-out}}$), we evaluated the log-likelihood, $\sum_{t \in \mathcal{T}_{\text{held-out}}} \{y_t \ln(\sigma(\hat{g}_\theta(\mathbf{x}_t))) + (1 - y_t) \ln(1 - \sigma(\hat{g}_\theta(\mathbf{x}_t)))\}$. We averaged the log-likelihood across all five folds and across all four cells, and chose the hyperparameter combination $\theta$ that yielded the largest average log-likelihood, resulting in a radial lengthscale of 8, a power lengthscale of 16, and a kernel amplitude of 8.

The GP parameters for generating the simulations in Figure 3, inferring the resulting ORFs, and generating synthetic optogenetics experiments with two and three spatial dimensions are given in Table S1. For reference, a typical ORF mean function is given in Figure S2.

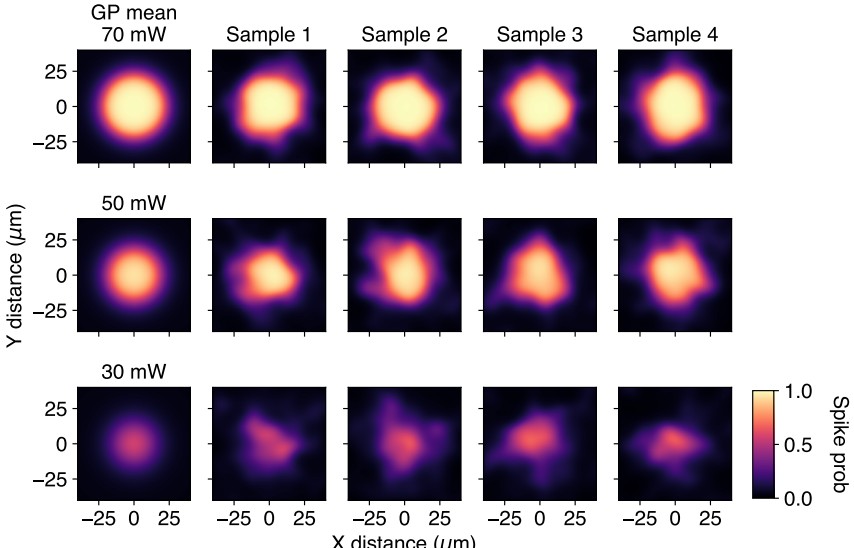

Figure S2: Example mean function (shown at three powers) used for simulations (left column). Also shown are four samples from the ORF prior corresponding to this mean function (right four columns). Parameters given in Table S1.

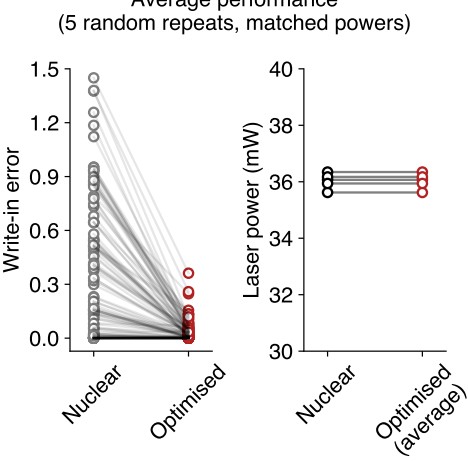

Figure S3: Performance of Bayesian target optimisation compared to nuclear stimulation when average laser powers are matched. Each repeat consists of randomly reassigning neurons different optogenetic receptive fields while keeping their spatial positions fixed, and performing a full mapping and target optimisation sequence. Within each repeat, nuclear stimulation was performed at the average optimised power. Left: average write-in error for nuclear stimulation remains substantially higher than with optimised stimulation despite having matched average powers. Each circle represents a single neuron. Right: powers used within each repetition. Note that these powers are lower than the 40 mW used to benchmark performance in Figures 4 and 5 in the main text.

| Parameter | Symbol | Value |
|---|---|---|
| Simulations (data generation) | | |
| Mean function excitability | $\rho$ | 0.125 |
| Mean function width | $\sigma_m^2$ | $3 \times 10^2 \ \mu$m |
| Spike threshold | $\theta$ | 3.5 |
| Kernel radial lengthscale | $\lambda_s$ | 8 $\mu$m |
| Kernel power lengthscale | $\lambda_I$ | 20 mW |
| Kernel amplitude | $\alpha^2$ | 0.2 |
| Kernel marginal variance | $\sigma_d^2$ | $10^{-5}$ |
| Simultaneously stimulated neurons during ORF mapping | $J$ | 10 |
| Simulations (ORF inference) | | |
| Mean function excitability | $\rho$ | 0.125 |
| Mean function width | $\sigma_m^2$ | $3 \times 10^2 \ \mu$m |
| Kernel radial lengthscale | $\lambda_s$ | 5 $\mu$m |
| Kernel power lengthscale | $\lambda_I$ | 16 mW |
| Kernel amplitude | $\alpha^2$ | 1 |
| Kernel marginal variance | $\sigma_d^2$ | $10^{-5}$ |
| Learning rate for spike thresholds ($\{\theta_n\}_{n=1}^N$) | $-$ | 5 |
| Number of random initialisations | $-$ | 5 |
| Synthetic optogenetics experiments (two spatial dimensions) | | |
| Mean function excitability | $\rho$ | 0.175 |
| Mean function width | $\sigma_m^2$ | $3 \times 10^2 \ \mu$m |
| Kernel radial lengthscale | $\lambda_s$ | 8 $\mu$m |
| Kernel power lengthscale | $\lambda_I$ | 16 mW |
| Kernel amplitude | $\alpha^2$ | 8 |
| Kernel marginal variance | $\sigma_d^2$ | $10^{-5}$ |
| Learning rate for spike thresholds ($\{\theta_n\}_{n=1}^N$) | $-$ | 5 |
| Number of random initialisations | $-$ | 5 |
| Synthetic optogenetics experiments (three spatial dimensions) | | |
| Mean function excitability | $\rho$ | 0.175 |
| Mean function width (x/y dimensions) | $\sigma_m^2$ | $3 \times 10^2 \ \mu$m |
| Mean function width (z dimension) | $-$ | $3 \times 10^3 \ \mu$m |
| Kernel radial lengthscale (x/y dimensions) | $\lambda_s$ | 8 $\mu$m |
| Kernel axial lengthscale (z dimension) | $\lambda_z$ | 32 $\mu$m |
| Kernel power lengthscale | $\lambda_I$ | 16 mW |
| Kernel amplitude | $\alpha^2$ | 8 |
| Kernel marginal variance | $\sigma_d^2$ | $10^{-5}$ |
| Learning rate for spike thresholds ($\{\theta_n\}_{n=1}^N$) | $-$ | 5 |
| Number of random initialisations | $-$ | 5 |

Table S1: Parameters used for simulations and generating synthetic optogenetics experiments.

## 6.8 Additional examples of optogenetic receptive fields from cell-attached recordings

Figures S4 to S7 show examples of four ORFs that have been comprehensively mapped using two-photon optogenetic stimulation and cell-attached recordings of evoked spikes. Note the unpredictable differences in ORF shape across laser powers and depths, motivating a nonparametric approach.

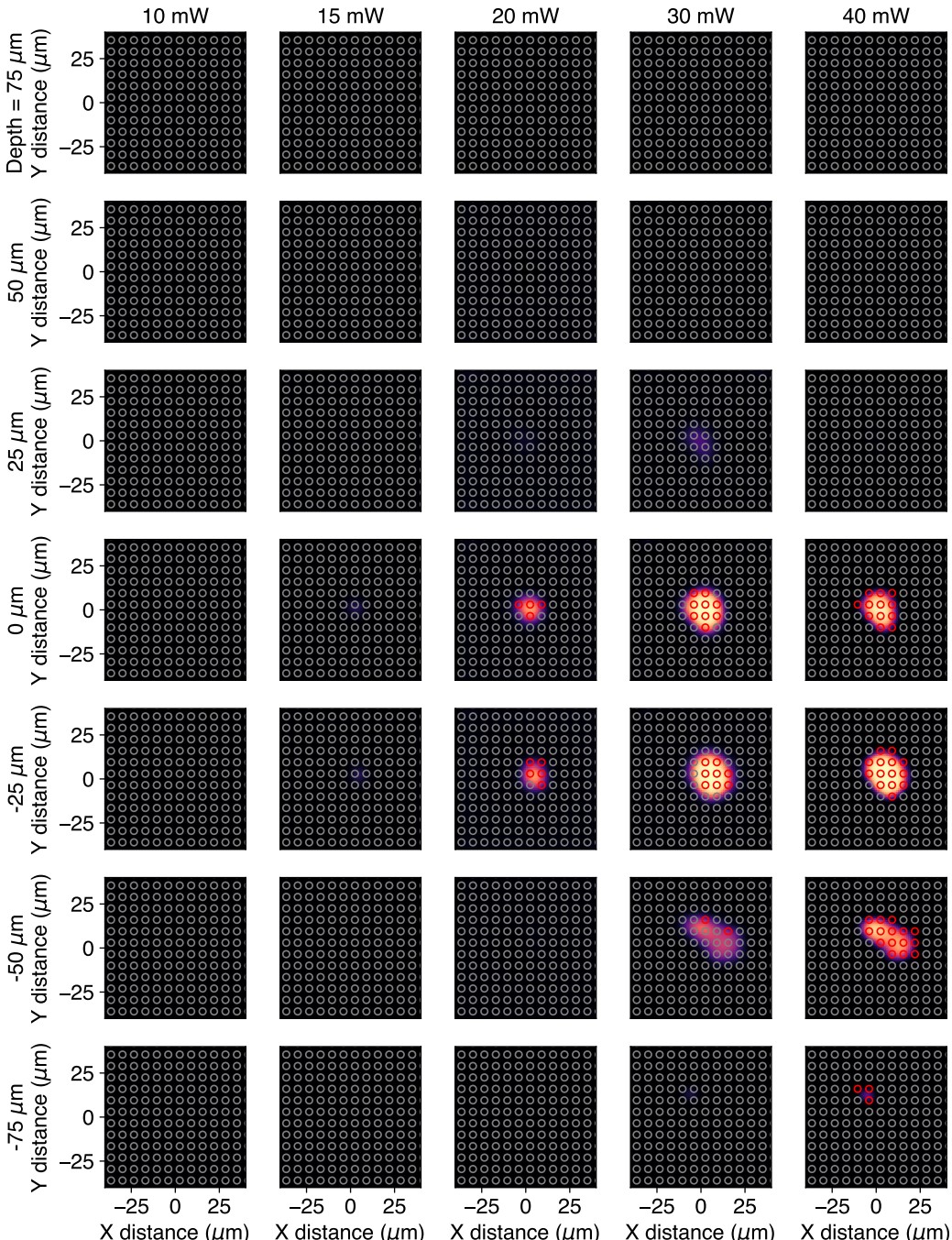

Figure S4: Loose-patch recording and inferred ORF (experiment 1/4).

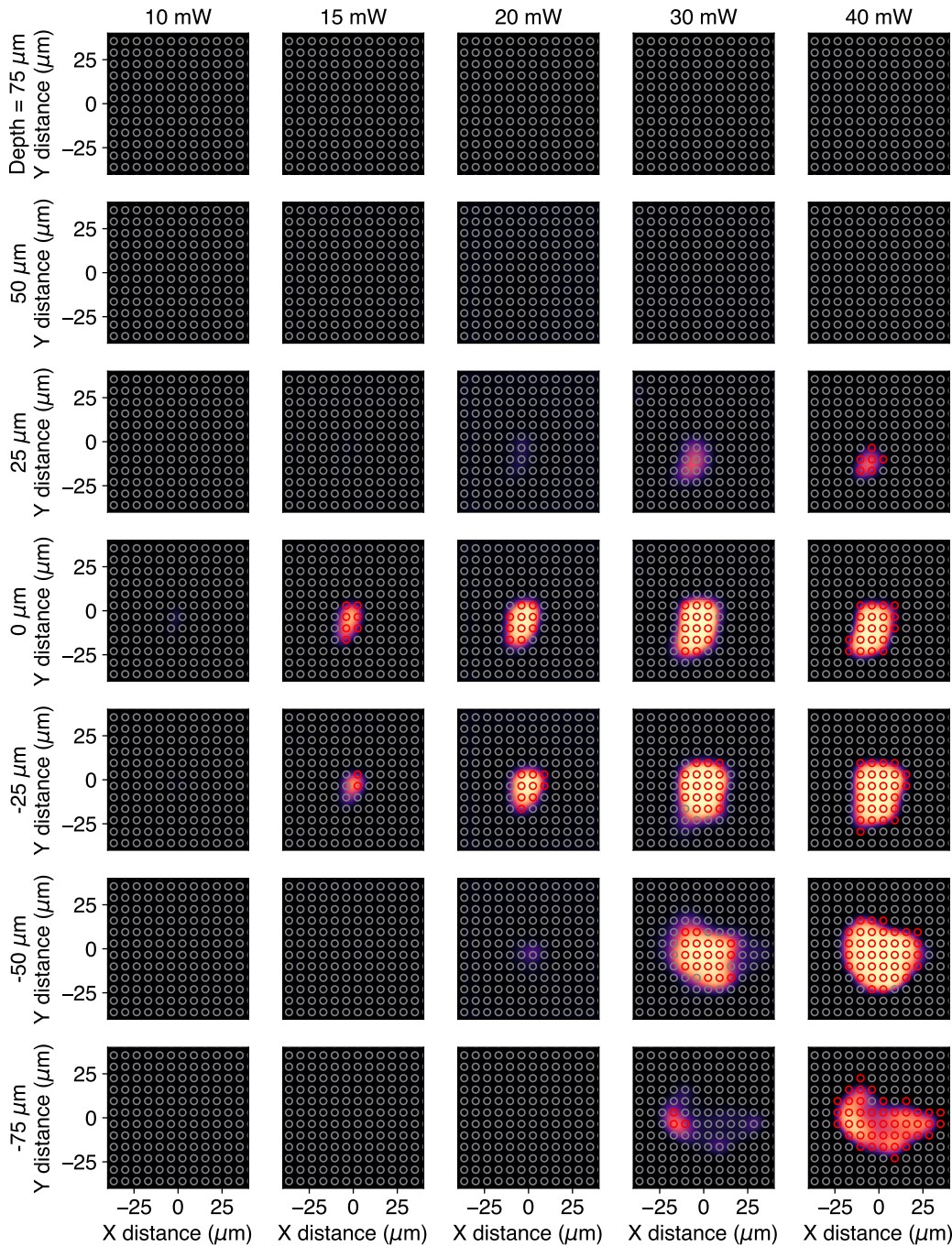

Figure S5: Loose-patch recording and inferred ORF (experiment 2/4)

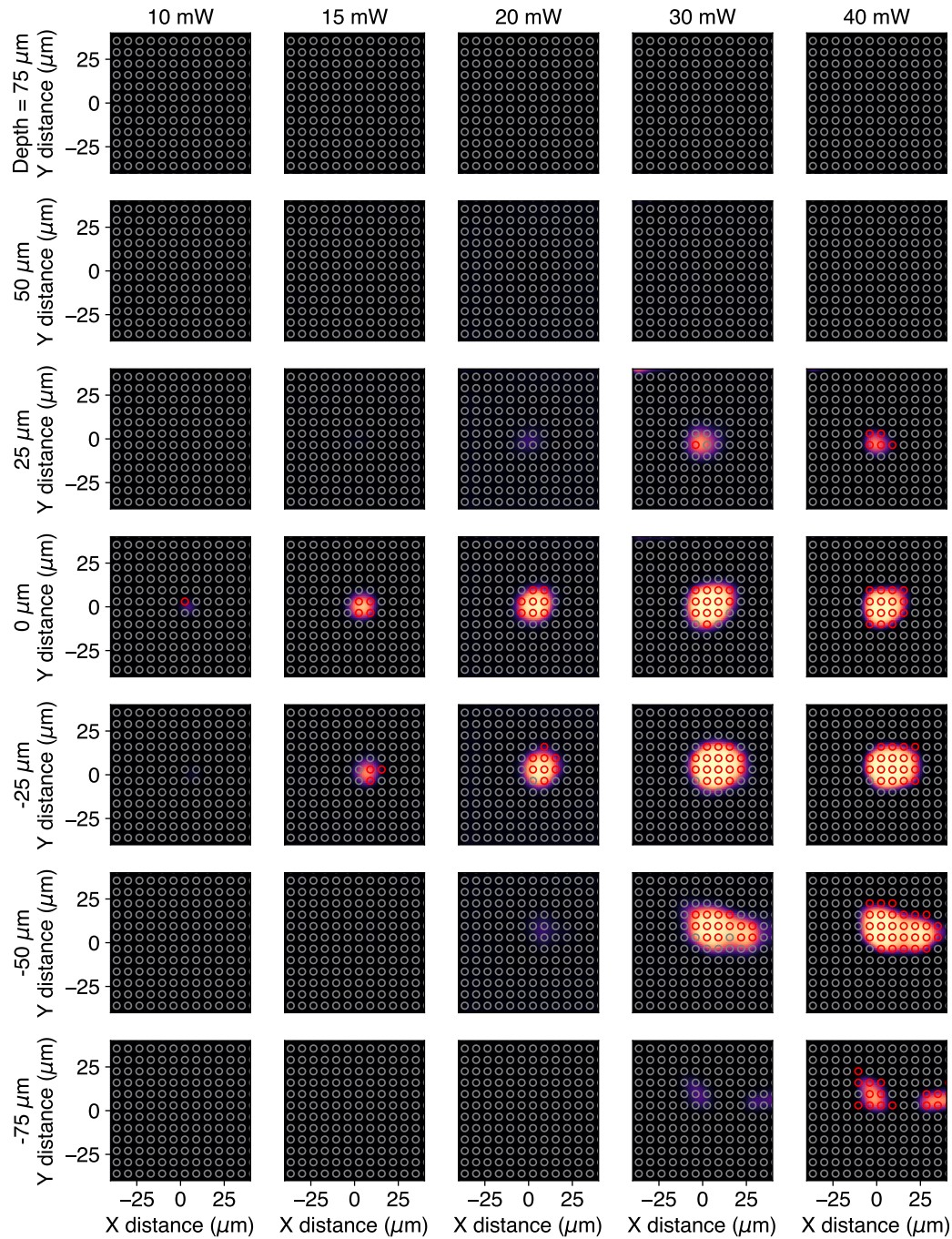

Figure S6: Loose-patch recording and inferred ORF (experiment 3/4)

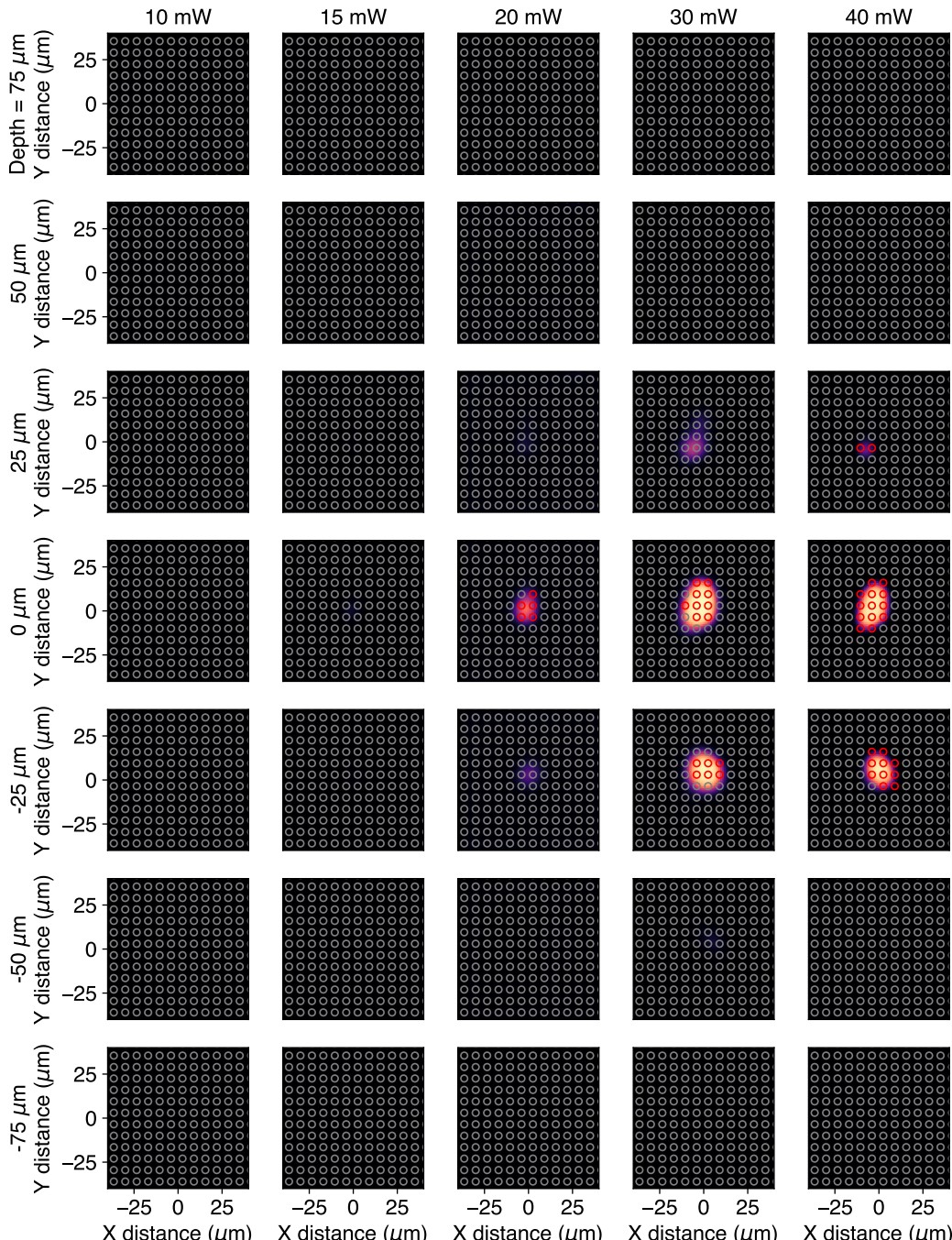

Figure S7: Loose-patch recording and inferred ORF (experiment 4/4)

