# OpenReview forum: "Bayesian target optimisation for high-precision holographic optogenetics"
_NeurIPS.cc/2023/Conference — NeurIPS 2023 spotlight_

### Official Review · Reviewer_G2K6 · 2023-06-28

**Soundness:** 4 excellent
**Presentation:** 4 excellent
**Contribution:** 3 good
**Rating:** 7
**Confidence:** 5

**Summary:**

This paper proposes a new method for limiting off-target optogenetic stimulation based on Gaussian Process modeling. In holographic photostimulation, the goal is to excite specific neurons via targeted laser light, but widespread expression of opsins may result in additional neurons not in the desired population ("off-target" neurons) also firing action potentials. The proposed method uses approximate GP inference (MAP estimation) in combination with a novel gradient-based optimization method to refine target locations in order to minimize the $L_2$ distance between the desired and evoked patterns of stimulation.

This is a very nice paper that combines good modeling with a very thoughtful approach to experimental realities. While direct experiments will be needed to fully test its efficacy, it stands to make an important contribution to this particular neuroscience application.

**Strengths:**

- Careful modeling of features of real experiments.
- Use of a scalable GP approach.
- Clear exposition.

**Weaknesses:**

- Much of the inference algorithms seems specific to some of the particular modeling choices made (particularly the need for a convex loss in Line 1 of Algorithm 1).
- Figs 4-5 should clearly be labeled as _simulation experiments_ based on real data; this is clearly explained in ll. 212-214 but elided in, e.g., ll. 254-56, which makes it seem as if the optimization was performed and validated as part of data collection.
- I was suprised not to see references to Fletcher and Rangan (2014) and Draelos and Pearson (2020) in the related work section.

**Questions:**

- Line 102 defines $\sigma$ as the sigmoid function. Does that mean logistic? (E.g, $\tanh$ is also sigmoid in shape but clearly not intended.)
- It is not stated in the paragraph surrounding Eq. 2 that $I$ is the laser power. It might help to simply state $\mathbf{x} = (c_1, c_2, I)$.
- Line 105 states that one should have $g_n(x) \ge 0$ so that stimulation should always have a positive effect, but in the supplement, this condition is waived in the single-target case. I assume this is because one can compensate in this case by taking $\theta = \inf_{x} g_n(x)$? If so, it might help to articulate this in the supplement.
- what is the MAP version of the ORF $\hat{g}_n$? This is just $\hat{g}_n(\mathbf{X})$, with $\mathbf{X}$ the matrix of test data points?
- Why use MSE and not cross-entropy, which would be the standard assumption for a desired set of observations under a Bernoulli model?

**Limitations:**

- Much of the inference algorithm is quite closely tied to the experimental setup. This is both a strength and a limitation, since the proposed work is somewhat unlikely to find broader applications.

---

> ### Author Rebuttal · Authors · 2023-08-09
>
> Thanks again for your helpful feedback.
>
> **Figs 4-5 should clearly be labeled as _simulation experiments_ based on real data**
>
> Thanks for noting this ambiguity. We will update the figure captions and lines 254-256 accordingly.
>
> **Missing references**
>
> We thank you for pointing out that we had not cited these papers, these were poor and unintended omissions on our part that we will fix. Notably, Draelos and Pearson (2020) is a recent inspiration for our work that we had absolutely meant to cite in the related work section.
>
> **Clarification of which sigmoid function is being used**
>
> Yes, we did indeed mean the logistic sigmoid and we will update the main text to clarify this.
>
> **Clarification of laser power parameter**
>
> Thanks for pointing out that this is unclear -- we will follow your suggestion!
>
> **Non-negativity is waived in the single-target case**
>
> When we waive the non-negativity constraint in the single-target case, we actually drop the threshold parameter entirely, much like in Gaussian process models of visual receptive fields [14]. This is because now if a point on the optogenetic receptive field (ORF) becomes inhibitory (by taking a negative value), it will not conflict with excitation from any other hologram, and therefore will still effectively model the response to optogenetic stimulation. We will clarify this in the supplement.
>
> **What is the MAP version of the ORF $\hat g_n$?**
>
> The MAP version of the ORF is $\hat g_n(\mathbf{X})$ (i.e. the ORF evaluated at $\mathbf{X}$), but where $\mathbf{X}$ is the set of "training" points probed during the ORF mapping phase (perhaps this is what you meant already). We will be explicit about this in section 3.2.
>
> **Why use MSE and not cross-entropy?**
>
> No particular reason, other than that it appeared to be an effective loss function in practice!
>
> Thanks again for all your comments.

---

> > ### Comment · Reviewer_G2K6 · 2023-08-10
> >
> > I appreciate the authors' thoughtful replies to my questions and congratulate them on a very nice result.

---

### Official Review · Reviewer_5v9K · 2023-06-30

**Soundness:** 3 good
**Presentation:** 4 excellent
**Contribution:** 4 excellent
**Rating:** 8
**Confidence:** 4

**Summary:**

The problem the authors tackle in their manuscript is the problem of target selection in holographic optogenetics. Briefly, of the many neurons in a field of view, many experiments require the selective stimulation of a small subset of cells, while minimizing off-target stimulation that may muddy the interpretation of the data. This question is of considerable importance to the field of systems neuroscience, and several experimental methods have been pursued trying to solve it, such as the use of soma-localized opsins or sparse expression of optogenetic protein. The authors present in this manuscript a complementary, Bayesian computational approach that takes into account differences in sensitivity of stimulation of different neurons in a given FOV to optimize laser intensity of position to minimize off-target stimulation while maintaining sufficient target stimulation. The authors use gaussian processes to GPs to infer the response properties of neurons based on stimulation parameters. The authors validate this approach in simulation and in real optogenetic experiments.

**Strengths:**

Originality: The authors introduce a novel computation algorithm to address a pressing issue in systems neuroscience; that of off-target stimulation. While the authors note that similar methods have been used to infer receptive fields for other types of stimulation, the authors cleverly employ these methods in a new way for inferring the responses of neurons to optogenetic stimulation to address a technical challenge facing the field. This software approach complements experimental approaches and can be readily combined with them. Similar computational approaches of thus far remained lacking, so the work is pioneering in this regard.

Clarity: The manuscript is clear with respect to how it poses the question being addressed, how it delineates its approach, and its statement of results.

Significance: The authors make a significant contribution to a timely question for systems neuroscience and will likely see significance use by labs performing holographic optogenetic stimulation, especially for applications in which minimizing off-target stimulation is crucial. The method could also see use in designing stimulation protocols to precisely pattern the activity of an ensemble of neurons, although that is a future direction.

**Weaknesses:**

Optimizing stimulation according to one parameter (in this case, minimization of off-target stimulation) often necessitates trade-off to other parameters. The authors bring up one such parameter, the time required to map the ORF and perform the optimization. While this consideration will vary from experimenter to experimenter, as the authors suggest, other considerations might not. For example, by moving stimulation locations off-center relative to the soma, stimulation reliability might become more sensitive to sample motion. Furthermore, the authors assume a ORF fixed during time; the manuscript would benefit from additional consideration for how the ORF changes with sustained stimulation, as desensitization of the channelrhodopsin could significant affect the ORF. In general, the authors should further explore the limitations and trade-offs of this method. This weakness is minor, however, since I assume that these other considerations could be incorporated as parameters to be optimized.

**Questions:**

Are the real-world validations provided using soma-localized channelrhodopsin, or is there channelrhodopsin in the processes? Does the method provide a greater benefit in one situation versus the other?

As stated in weaknesses section:
How sensitive is an OTS-minimized stimulation protocol to sample motion? How does this compare to soma-targeting? How about to changes in neuron sensitivity to stimulation? Can these be incorporated into optimization?

**Limitations:**

The authors have adequately addressed the limitations.

---

> ### Author Rebuttal · Authors · 2023-08-09
>
> Thank you again for spending the time to review our submission and for providing your comments!
>
> **Sensitivity to sample motion**
>
> Thanks for raising this relevant point. It is true that sample motion could affect the optimality of the computationally identified stimuli. In practice, we would therefore recommend performing online motion correction [11] immediately before stimulation, so that the learned optogenetic receptive field (ORF) posterior is as closely aligned in space to the true "real-time" receptive field as possible.
>
> **Sensitivity to opsin nonstationarity**
>
> Desensitisation of opsins following prolonged illumination has indeed been observed [12]. However, after the ORFs have been mapped and holographic stimuli have been optimised, it is straightforward to periodically recompute the posterior as the experiment proceeds. This would provide an updated estimate of the laser power required to maintain optimal precision. Note that this does not require redoing the whole ORF mapping phase as we would not expect the shape of the ORF to change drastically beyond desensitisation.
>
> We will update the manuscript to discuss these important points.
>
> **Are the real-world validations provided using soma-localized channelrhodopsin?**
>
> Yes, the validation using experimental data is performed with ChroME2f, a state-of-the-art soma-targeted opsin [13].
>
> **Does the method provide a greater benefit in one situation versus the other?**
>
> We have not explored applying our technique in the case of an opsin that is not localised to the soma, because in such experiments the precision of any optogenetic manipulation is extremely poor [7, 8]. We therefore recommend that any experimenter wanting to achieve single-cell precision first change to a soma-localised opsin before attempting any computational optimisation.
>
> Thanks again for your comments.

---

> > ### Comment · Reviewer_5v9K · 2023-08-10
> >
> > Thank you for your reply! My opinion that the paper is technically solid and impactful remains, as does my score.

---

### Official Review · Reviewer_4tSm · 2023-07-06

**Soundness:** 3 good
**Presentation:** 4 excellent
**Contribution:** 3 good
**Rating:** 7
**Confidence:** 4

**Summary:**

The authors present a method for reducing off-target stimulation of neurons during photostimulation by modifying the laser power and target locations. They use a Bayesian optimization approach to determine neuron responses (ORFs) to stimulations at different targets and laser powers and then choose the optimal target parameters to attain a desired neural response to stimulation. In contrast to typical stimulation targeted to the desired neuron’s soma, stimulating somewhat ‘off-target’ can still produce spikes in the desired neurons and avoid stimulating its nearby neighbors.

**Strengths:**

Originality: This work is well-motivated by the growing number of methods used for direct neuronal stimulation using photostimulation to determine e.g. causal effects of single neuron activity or the functional structure of a neural circuit. Here, rather than address the inference of neural function or connectivity itself, the authors present a solution to the problem of off-target stimulation (OTS), an experimental reality where neural targets not selected for stimulation are stimulated by the laser anyway (due to e.g. poor spatial resolution). The idea of using Bayesian optimization (BO) and finding locations for off-soma stimulations that reduce OTS is very appealing and the method proposed here with BO, GPs to model neural responses, and optimization of stimuli location shows a nice original combination of techniques to address this problem.

Quality: The methods used (GPs, BO, gradient descent for optimization) are all well-established and robust. The inference method for the ORFs with the additional non-negativity constraint is nicely explained. The results clearly demonstrate an advantage of this method over traditional nuclear stimulation for the single-neural-target case in particular.

Clarity: Overall the paper is very well written. Ideas from the literature are well-sourced and the authors make it clear how their approach uses some similar techniques and where their method is novel. The figures are good quality as well. The code is relatively clear, though it could use additional documentation or commenting for ease of use.

Significance: As more neuroscience researchers turn to photostimulation for causal testing of hypotheses, this method should prove useful for the cases where laser accuracy is insufficient for selective stimulations.

**Weaknesses:**

One weakness is the lack of experimental data, though it is understandably still difficult to obtain. While the authors mentioned slice data, this was only used as input to their simulations. Looking at the various ORFs mentioned in the supplement, is it reasonable to think that only 4 types of ORFs in the simulation is enough? And if these ORFs that were experimentally obtained were done so via stimulating a grid nearby, would that change the estimation of 1 neuron's ORF if another neuron nearby was also being stimulated (in an ensemble)? In vivo, one would expect that confounding neural activity could occur due to connectivity in addition to laser power spillover.

It is unclear if, at the end, this method is useful for ensemble stimulation in addition to single target paradigms. While the initial motivation and explanation of the method appears to be geared towards both cases (including Algorithm 1), it is not clear if in Figures 4-5 the optimization of each 'single-target hologram' is the end result of an optimization across multiple targets, or if the optimization was made per target. The different factors examined here ('density', size of ensemble, xy plus z dimensions) are all nicely presented but showing the method works in the 'worst case' (high density, larger ensembles, fully 3D, ...) would make this work stronger.

**Questions:**

1. How well does this scale from an optimization (time) perspective? Can this be done for thousands of neurons? Ensembles of 50?

2. How does the exact neuron expression matter (to the ORF shape)? Can a simplified version be used during the 2nd step of target optimization?

3. Similarly, what do ORFs look like? Are there other references used for establishing the prior beyond the few shown here?

4. Would it be possible to integrate spatial information (from e.g. calcium imaging) into the ORF prior in a useful way? To minimize the number of required mapping stimulations before the target optimization step.

5. It appears that the method uses random locations to first map the ORFs. Would it be possible to optimize the selection of locations for the ORF mapping based on the desired targets for later stimulation?

6. If Algorithm 1 needs to be run many times with different seeds, how long does it take to determine the optimal stimulation pattern? How practical is this in an experimental setup?

7. Clarification on Figure 3: Line 199 says increasing density of neurons, and Fig. 3a, shows 50 - 150 neurons (in the same 250 x 250 um plane) – would be nice to explicitly state that is what was done in simulation, and state what the density therefore is. Why were these density ranges chosen (for different example brain regions)? Are the number of neuron-neuron connections a potential factor here?

8. Is the performance increase the same for lower laser power in the nuclear condition? Ie if we used nuclear stimulation at lower power would we do just as well as optimized? Figure 4 & 5 indicate the optimization chose overall lower laser powers than initial settings.


**Limitations:**

The authors discussed some trade-offs between the time spent mapping the ORFs and the need for single-cell precision stimulations. They mention potential future work to minimize the time needed for Bayesian target optimization, but fail to detail exact time requirements for the current method.

---

> ### Author Rebuttal · Authors · 2023-08-09
>
> Thanks very much for your insightful comments and feedback! Unfortunately we had to cut much of our response to meet the character limit -- we would have liked to address every point and with more detail.
>
> **Is it reasonable to think that 4 ORFs are enough?**
>
> We do not have a sense of the variability of optogenetic receptive field (ORF) shapes beyond the slice recordings that have been made. However, we have attempted to account for this by sampling over many different GPs in our simulations, which should extensively explore the space of ORFs.
>
> **It is not clear if in Figures 4-5 the optimization of each single-target hologram is the end result of an optimization across multiple targets, or if the optimization was made per target**
>
> Thank you for highlighting this ambiguity. The optimisation was made for each target individually, and we will further clarify this in the figure caption and main text.
>
> **How does this scale? Can this be done for thousands of neurons? Ensembles of 50?**
>
> Thank you for raising this important point. We first note that experiments of such scale (with thousands of neurons being stimulated) are beyond what we expect the current implementation of Bayesian target optimisation can handle. Such scales could become feasible computationally if more advanced GP estimation techniques were employed, though it's not clear to us that biological experiments of this size (where a single SLM can flexibly deliver 2p excitation to thousands of neurons individually, within a single stimulation field, and with high spatiotemporal fidelity) are feasible with existing technology.
>
> While our objective in this submission was to establish the feasibility of overcoming off-target stimulation computationally rather than achieving the fastest possible implementation, we have provided some preliminary runtime data in the attached supplement (see **Table 1** in the rebuttal pdf), and we plan to extend the characterisation for the camera-ready copy.
>
> Also note that both ORF inference and target optimisation are performed sequentially, and thus major speed gains could be made by parallelising ORF inference over different neurons, and target optimisation over different desired ensembles.
>
> The runtimes given in **Table 1** demonstrate suitability for typical all-optical experiments using GCaMP6 indicators [6], the most widely used class of calcium sensors. We believe there is room for further optimisation (e.g. with more parallelisation), and in future work we plan to push the efficiency of the technique for use with the class of faster GCaMP8 indicators. We hope this also answers Question 6 from the reviewer, which we do not have space to respond to in a separate comment.
>
> **How does the exact neuron expression matter to the ORF shape? Can a simplified version be used during the 2nd step of target optimization?**
>
> While at a fixed power and in two dimensions some ORFs might be reasonably well-described by a simpler parametric model (e.g. a circle), in three dimensions the shape of the ORF can change so drastically and with unpredictable asymmetries that a single parametric model is unlikely to capture their varying shape with much accuracy at all (see supplemental figures S3-S6). Thus, we believe we have taken the simplest nonparametric approach (an RBF Gaussian process) that adequately describes the data.
>
> During the second step of target optimisation, a simple parametric ORF model might go some way in better repositioning holograms compared to naive nuclear stimulation, but for achieving the highest performance possible our experiments have found that it is certainly advantageous to exploit asymmetries in the ORF shapes, especially in three dimensions. For example, with the neuron shown in supplemental figure S4, it is clearly better to learn during an experiment that one should stimulate this neuron deeper in the tissue (if trying to avoid other neurons) than shallower.
>
> **Are there other references used for establishing priors beyond the few shown here?**
>
> The convention in the two-photon optogenetics field has been to characterise the ability to stimulate a neuron by moving only in one dimension at a time (radially or axially, i.e. "left and right" or "up and down"; see e.g. [7-9]). The only existing reference of ORFs profiled at this level of detail (i.e. a complete grid) is [10, Figure 1G], which is similar data to what we present in the supplement.
>
> **Would it be possible to integrate spatial information (from e.g. calcium imaging) into the ORF prior?**
>
> We thank the reviewer for making this interesting suggestion. In most cases, calcium sensors and opsins are expressed through separate viral vectors, and neurons are not guaranteed to express both proteins at once. However, some research groups have developed constructs that fuse the two proteins together directly [10], in which case the relative brightness of a calcium transient across the cell soma and proximal dendrites might provide some hint at different locations that the cell could be better activated, though it remains to be seen how useful this would actually be in practice.
>
> **Would it be possible to optimize the selection of locations for the ORF mapping based on the desired targets for later stimulation?**
>
> Yes! We definitely think this can and should be done in practice. If one only wishes to stimulate a desired subset of neurons, then ORF mapping should only be performed in neighbourhoods surrounding those neurons.
>
> **If we used nuclear stimulation at lower power would we do just as well as optimized?**
>
> We performed a control analysis by matching the nuclear stimulation laser power to the average optimised laser power (see **Figure 1** in the rebuttal supplement) and reran the mapping and target optimisation phases multiple times while randomly reassigning neurons different ORFs. This process showed that we maintain a very similar improvement in performance as what was shown in the main text.
>
> Thanks again!

---

> > ### Comment · Reviewer_4tSm · 2023-08-12
> >
> > Thanks to the authors for their extensive replies to my and other reviews. I have read over the other reviews and the author responses, and think the paper is strengthened by the improvements the authors will make.

---

### Official Review · Reviewer_FU6t · 2023-07-07

**Soundness:** 3 good
**Presentation:** 3 good
**Contribution:** 2 fair
**Rating:** 6
**Confidence:** 4

**Summary:**

The authors present a set of methods to efficiently characterize the activation field under optogenetics and optimize the optogenetic stimulation patterns to target certain cells while avoiding the others.

**Strengths:**

The paper tackles an important problem for reproducing the neural responses with high resolution in a neural circuit. The presented approach is direct, principled and effective.

**Weaknesses:**

 The experimental validation of the method is limited. It is unclear how the real-data was used in analysis (Sec 4.2), and if it represents what would happen in a real experiment.

**Questions:**

• How is linear summation of activation from different optogenetic stimulation sites justified, especially if the sites are very close to each other?
• Could a simple stimulation optimization approach, where the location is at the intersection of high g(x) for one cell and low g(x) for other cells, work?

A characterization of the efficiency (reduction in the number of measurements for the same estimation quality) is missing.

**Limitations:**

Limitations have been adequately addressed.

---

> ### Author Rebuttal · Authors · 2023-08-09
>
> Thanks again for spending the time to review our submission and for providing your comments.
>
> **The experimental validation of the method is limited. It is unclear how real-data was used in analysis (Sec 4.2) and if it represents what would happen in a real experiment.**
>
> Thanks for noting that there is room to improve the clarity in the text about how the real data was used in the analysis. As we commented on in the global rebuttal, the ideal experimental data for validating these techniques are not currently available to us (i.e., an _in vivo_ demonstration of the technique, though we are actively collaborating to acquire such data). Therefore, we are limited to working with the data that we have, which come from slice experiments.
>
> We made use of two kinds of experimental data: (1) detailed optogenetic receptive field (ORF) maps for single neurons, obtained by making a loose-patch recording of a single neuron, stimulating at regularly spaced locations surrounding the patched neuron and at multiple powers, and then fitting the GP-Bernoulli model; and (2) a fluorescence image of opsin expression from a separate experiment showing a typical distribution of opsin-expressing neurons in space. To use these data to simulate an experiment, we randomly assigned each opsin-expressing neuron in (2) an ORF from (1), and used these ORFs to sample responses to neural stimulation at arbitrary locations and powers.
>
> **How is linear summation from multiple nearby sites justified?**
>
> To our knowledge, an experimental characterisation of the photocurrent evoked by multiple closely positioned holograms has not yet been performed. However, the photocurrent directly depends on the number of opsin molecules illuminated [4], and we do not expect an interaction effect between two-photon laser pulses that would nonlinearly recruit opsin molecules beyond more quickly approaching a saturation point (that is currently accounted for by the sigmoid nonlinearity).
>
> That being said, some prior experiments using two-photon glutamate uncaging have found cases of nonlinear summation when stimulating multiple (_distal_) dendritic spines on the same branch [5]. However, in the experiments that our technique is designed for, opsin expression is restricted to the soma and (at most) the _proximal_ dendrites. We expect that instances of simultaneously stimulating multiple points along the same segment of proximal dendrite immediately adjacent to a target neuron's soma will be very rare, and we do not currently know whether this would measurably change the ability to optogenetically stimulate the neuron.
>
> **What about a simpler approach of finding high $g(x)$ for one cell and low $g(x)$ for other cells?**
>
> We thank the reviewer for this intuitive suggestion, and note that for the single-target case this is similar to what the proposed gradient descent solution finds (though the supplementary algorithm for single-target optimisation further accounts for posterior uncertainty in the inferred ORFs). However, this does not generalise to the ensemble stimulation case because it does not account for the contribution from multiple holograms at once.
>
> **A characterisation of the efficiency is missing**
>
> While we show the effect of probing the ORFs with fewer stimuli (i.e. using downsampled grids) in the supplement, we will provide further data related to statistical efficiency in the final version of the manuscript and make an explicit reference to the result in the main text.
>
> Thanks again for your comments!

---

### Author Rebuttal · Authors · 2023-08-09

Thank you all for your excellent feedback and positive evaluation of our submission! We presented Bayesian target optimisation, a computational approach to overcoming off-target stimulation in two-photon optogenetics experiments. We are delighted that every reviewer clearly understood the motivation, methodological contributions, and experimental relevance of our work.

Below we consider the comments that are specific to each review, but first we would like to address the fact that multiple reviewers had concerns regarding experimental data. In most cases, the ideal data for calibrating the internal model components unfortunately do not yet exist or are not currently available to us. We are actively working with experimental collaborators to validate our technique _in vivo_. However, obtaining new experimental data to address the specific concerns of individual reviewers is (hopefully understandably) beyond the scope of this NeurIPS submission. We have therefore done our best to provide, where possible, additional references to what is known in the literature. Nevertheless, we believe that our chosen validation approach (combining data from separate real experiments) comes as close to a direct experimental validation as is currently feasible, and is more closely tied to real experimental data than any existing computational work at NeurIPS in this area [1-3].

Please note that we have attached some additional data in response to Reviewer 4tSm, and have collected all references in this general rebuttal in order to meet character limits. Thanks again all for your work in reviewing our paper.

**References**

1. Hu, T., & Chklovskii, D. (2009). Reconstruction of sparse circuits using multi-neuronal excitation (RESCUME). Advances in Neural Information Processing Systems, 22
2. Shababo, B., Paige, B., Pakman, A., & Paninski, L. (2013). Bayesian inference and online experimental design for mapping neural microcircuits. Advances in Neural Information Processing Systems, 26.
3. Draelos, A., & Pearson, J. (2020). Online neural connectivity estimation with noisy group testing. Advances in Neural Information Processing Systems, 33, 7437-7448.
4. Rickgauer, J. P., & Tank, D. W. (2009). Two-photon excitation of channelrhodopsin-2 at saturation. Proceedings of the National Academy of Sciences, 106(35), 15025-15030.
5. Losonczy, A., & Magee, J. C. (2006). Integrative properties of radial oblique dendrites in hippocampal CA1 pyramidal neurons. Neuron, 50(2), 291-307.
6. Russell, L. E., Dalgleish, H. W., Nutbrown, R., Gauld, O. M., Herrmann, D., Fişek, M., ... & Häusser, M. (2022). All-optical interrogation of neural circuits in behaving mice. Nature Protocols, 17(7), 1579-1620.
7. Baker, C. A., Elyada, Y. M., Parra, A., & Bolton, M. M. (2016). Cellular resolution circuit mapping with temporal-focused excitation of soma-targeted channelrhodopsin. Elife, 5, e14193.
8. Shemesh, O. A., Tanese, D., Zampini, V., Linghu, C., Piatkevich, K., Ronzitti, E., ... & Emiliani, V. (2017). Temporally precise single-cell-resolution optogenetics. Nature neuroscience, 20(12), 1796-1806.
9. Mardinly, A. R., Oldenburg, I. A., Pégard, N. C., Sridharan, S., Lyall, E. H., Chesnov, K., ... & Adesnik, H. (2018). Precise multimodal optical control of neural ensemble activity. Nature neuroscience, 21(6), 881-893.
10. Bounds, H. A., Sadahiro, M., Hendricks, W. D., Gajowa, M., Gopakumar, K., Quintana, D., ... & Adesnik, H. (2023). All-optical recreation of naturalistic neural activity with a multifunctional transgenic reporter mouse. Cell Reports, 42(8).
11. Pnevmatikakis, E. A., & Giovannucci, A. (2017). NoRMCorre: An online algorithm for piecewise rigid motion correction of calcium imaging data. Journal of neuroscience methods, 291, 83-94.
12. Marshel, J. H., Kim, Y. S., Machado, T. A., Quirin, S., Benson, B., Kadmon, J., ... & Deisseroth, K. (2019). Cortical layer–specific critical dynamics triggering perception. Science, 365(6453), eaaw5202.
13. Sridharan, S., Gajowa, M. A., Ogando, M. B., Jagadisan, U. K., Abdeladim, L., Sadahiro, M., ... & Adesnik, H. (2022). High-performance microbial opsins for spatially and temporally precise perturbations of large neuronal networks. Neuron, 110(7), 1139-1155.
14. Park, M., Horwitz, G., & Pillow, J. (2011). Active learning of neural response functions with Gaussian processes. Advances in neural information processing systems, 24.

---

### Decision · Program_Chairs · 2023-09-21

**Decision:**

Accept (spotlight)

**Comment:**

This work aims to identify the use of Bayesian optimization to develop a targeting approach that minimizes the stimulation of non-target neurons that can occur during in-vivo neural stimulation using optical methods. The methods itself is sound, and the reviewers noted that the exploration is timely and complements well the development of optical methods in systems neuroscience.

The main weakness of this work is the difficulty in knowing just how well the method, with the approximations and modeling choices of the light-neuron interactions, will work in real datasets. The example data uses primarily simulated data (either toy data or sampling from in-vivo data). Granted that real data are difficult to obtain, there is more realistic simulation software available (Song et al. 2022 J. Neuo Methods), and simple characterizations like showing that the offsets optimized by the system are useful even under in vivo motion conditions, which can throw off targeting quite a bit. Regardless the reviewers considered the benefits of the work to outweight the shortcomings and so I recommend this work be accepted.